# LARGE-SCALE DYNAMIC GRAPH GENERATION VIA LLM-BASED AGENT SIMULATION

## ABSTRACT

Graph generation is a fundamental task that has been extensively studied in social, technological, and scientific analysis. For modeling the dynamic graph evolution process, traditional rule-based methods struggle to capture community structures within graphs, while deep learning methods only focus on fitting training graphs. This limits existing graph generators to producing graphs that adhere to predefined rules or closely resemble training datasets, achieving poor performance in dynamic graph generation. Given that graphs are abstract representations arising from pairwise interactions in human activities, a realistic simulation of human-wise interaction could provide deeper insights into the graph evolution mechanism. With the increasing recognition of large language models (LLMs) in simulating human behavior, we introduce GraphAgent-Generator (GAG), a novel simulation-based framework for dynamic text-attributed graph generation. Without training or fine-tuning process of LLM, our framework effectively replicates seven macro-level structural characteristics in established network science theories while surpassing existing baselines in graph expansion tasks by 11% on specific evaluation metrics. Through node classification task, we validate GAG effectively captures the intricate text-structure correlations in graph generation. Furthermore, GAG supports generating graphs with up to nearly 100,000 nodes or 10 million edges through large-scale LLM-based agent simulation with parallel acceleration, achieving a minimum speed-up of 90.4%. The source code is available at https://anonymous.4open.science/r/GraphAgent-2206.

## 1 INTRODUCTION

Graphs are mathematical structures representing pairwise interactions between entities through nodes and edges, serving as a fundamental concept in network science. They are widely used to model interaction behaviors across various domains, including social analysis (Fan et al., 2019), parallel computing (Hendrickson & Kolda, 2000), and program synthesis (Nguyen et al., 2012). A longstanding task in network science is graph generation. Given an observed graph dataset, researchers extract the underlying generative mechanisms to create models that can scale these observations into larger graphs. Network science theories constitute macro properties such as power-law degree distribution (Clauset et al., 2009). In contrast, micro properties include graph structure metrics like degree distribution and clustering coefficient (Martinkus et al., 2022). By comparing the macro and micro properties of the generated and real-world graphs, researchers gain deeper insights into graph evolution mechanisms.

Existing graph generation methods can be categorized into two types: (1) Rule-based methods, which rely on preset rules to generate graphs (Erdos et al., 1960; Barabási & Albert, 1999). These methods are designed to capture specific macro properties observed in real-world networks. However, the need for tailored models to capture each property complicates the integration of these methods into a unified framework. Additionally, they fall short in capturing micro-level properties in dynamic graph generation (Bergmeister et al., 2024). (2) Deep learning-based methods, which leverage self-supervised learning to capture graph structures. These methods mainly include auto-regressive methods (You et al., 2018) and one-shot methods (Vignac et al., 2023; Bergmeister et al., 2024; Simonovsky & Komodakis, 2018). While these techniques excel in fitting micro properties of observed graphs, they face challenges when generating larger graphs beyond the size of the observed dataset (Bergmeister et al., 2024) and struggle to capture macro properties in dynamic graph generation.

The limitations of previous methods stem from their attempts to use a single model to represent all forms of entity-wise interaction process. Instead of merely adhering to preset rules or fitting training data, a desirable graph generator understands how graphs are formed to generate structures that align with the underlying physical interaction process. For instance, the dynamics of human social interactions significantly influence the social network evolution Fowler & Christakis (2010). Fortunately, the emergence of LLMs like LLaMA (AI@Meta, 2024) and GPT-4 (OpenAI, 2023) has opened new avenues for graph generation. With their ability for human-like reasoning, LLM-based agents can effectively simulate complex interaction processes in human activities (Park et al., 2023).

In this work, we introduce GraphAgent-Generator (GAG), a novel human behavior simulation-based framework via LLM-based agents for graph generation. We propose the S-RAG algorithm to model the human interaction process with carefully designed LLM-based agents for human behavior simulation. Through continuous simulations, we can extract diverse graphs as abstract representations of collected interaction data. Furthermore, we propose N-ACTOR to accelerate the simulation process by parallel processing. Through experiments conducted in GAG, our main contributions are:

**(1) Graphs of Real-World Network Structures:** The generated graphs exhibit seven essential structural characteristics observed in real-world networks, including *power-law degree distribution*, *small-world*, *shrinking diameter* and etc. Specifically, GAG surpasses the best-performing baseline by 11% on specific evaluation metrics for graph expansion tasks.

**(2) Interpretability in Graph Generation:** To our knowledge, GAG is the first framework to generate text-rich data for modeling graph evolution, offering clear process interpretability. In the context of node classification using Graph Neural Networks (GNNs), the graph generated by GAG retains accuracy comparable to that of the real-world graph. This demonstrates the capability of GAG to produce effective text features for graphs.

**(3) Graph Generation via Large-Scale LLM-based Agent Simulation:** The framework supports the generation of graphs across ten distinct types, accommodating up to 10 million edges or nearly 100,000 nodes through simulations with up to nearly 100,000 LLM-based agents. Additionally, the N-ACTOR component accelerates the simulation process through parallel computing with a minimum speed-up of 90.4%.

## 2 RELATED WORK

**Graph Generation** As an extensively explored foundational task, existing graph generation methods mainly fall into two categories: (1) Rule-based methods, which gradually add nodes based on random (Erdos et al., 1960) or preferential attachment (Barabási & Albert, 1999) rules; Though (Barabási & Albert, 1999) successfully models power-law degree distribution, they struggle to capture the community structures prevalent in real-world networks (You et al., 2018). (2) Deep Learning based methods: In recent years, deep learning offer a new approach, aiming to capture the complex and diverse structures of real-world networks, which mainly fall into two categories: Autoregressive methods (You et al., 2018; Dai et al., 2020; Bergmeister et al., 2024) predict edges incrementally for each new node, while one-shot methods (Simonovsky & Komodakis, 2018; De Cao & Kipf, 2018; Liu et al., 2019; Vignac et al., 2023) generate entire graphs in a single step. However, these methods require large-scale training data and struggle to generate graphs outside the training distribution. Although some progress is made with extrapolating to out-of-distribution graphs (Bergmeister et al., 2024; Limnios et al., 2023), the maximum size of graph is limited to thousands of nodes.

**LLM-based Human Behavior Simulation** With LLMs demonstrating advanced capabilities in human-like responses and autonomous planning (Gao et al., 2023), they are increasingly recognized as a new paradigm for simulations across fields such as education (Chen et al., 2024), social dynamics (Park et al., 2023), and economics (Li et al., 2024b). In graph generation, De Marzo et al. (2023) first explores the scale-free property of power-law distributions in LLM-based agent interactions. Subsequently, Papachristou & Yuan (2024); Chang et al. (2024) examines additional social network properties. However, these simulations often lack realism due to simplified modeling of human behavior, such as name selection, and are typically constrained to fewer than 100 agents. Recently, Pan et al. (2024) introduced AgentScope, a framework enabling large-scale multi-agent simulations for simplified human behavior, demonstrated through number-guessing games. Building on this, we have enhanced AgentScope to simulate more complex human behaviors at a large scale.

## 3 THE GAG FRAMEWORK

In this section, we present GAG, a versatile LLM-simulation-based framework designed for large-scale graph generation. GAG aims to eliminate preset rules and training processes in graph generation through simulation-based methods.

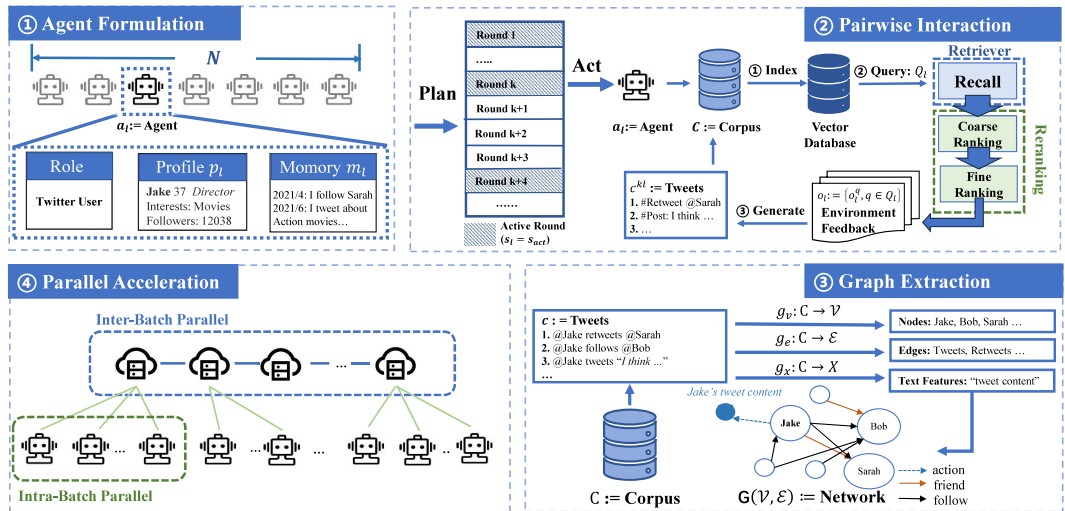

Figure 1: An illustration of the GAG Framework for generating social networks: (1) **Agent Formulation**, where diverse agents are initialized; (2) **Pairwise Interaction**, where agents interact by generating and retrieving content from a shared corpus; (3) **Graph Extraction**, where interactions are translated into a dynamic network of nodes and edges; and (4) **Parallel Acceleration**, which implements inter- and intra-batch parallelism to speed up the simulation process.

### 3.1 PROBLEM SETUP

Graphs serve as mathematical structures that represent relationships and interactions among entities (Bergmeister et al., 2024-05). In this paper, we consider dynamic graph generation for text-attributed graphs. In the graph generation task, we consider a seed graph $G(\mathcal{V}, \mathcal{E}, \mathcal{X})$, where $\mathcal{V} = \{v_i\}_{i=1}^{n}$ represents a set of nodes, $\mathcal{E} = \{e_{ij}\}$ denotes the set of edges connecting these nodes, and $\mathcal{X} = \{x_1, \ldots, x_n\}$ comprises the text attribute values associated with each vertex. Here, $x_i$ indicates the text attribute linked to node $v_i$. The graph generation model seeks to learn how $G$ evolves into a significantly larger graph $G'(\mathcal{V}', \mathcal{E}', \mathcal{X}')$, where $n' \gg n$ and $m' \gg m$, over time. The objective is for the generated graph $G'$ to accurately mirror the macro patterns of the original graph during evolution while maintaining its essential structural properties. In citation networks, for example, graph evolution closely reflects the temporal interactions involved in paper-citing behavior among authors. By simulating these interactions to capture the dynamics of graph evolution, we gain a deeper understanding of how human behavior patterns drive changes in graph structure.

To design an effective dynamic graph generator, we develop the GAG framework to simulate pairwise interactions in human activities, modeling graph evolution as a process of collecting interaction data repeatedly. Leveraging the ability of LLMs to mimic human-like behaviors (Li et al., 2023; Park et al., 2022; 2023), we construct LLM-based agents for human behavior simulation, denoted as $A = \{a_1, a_2, \ldots, a_N\}$, where $N$ is the number of agents. Each interaction process between $A$ corresponds to an update of the $G$. In the $k$-th simulation round, agent $a_l \in A$ observes the data from $G$ and generate a subgraph $G_k^l$. Then, the subgraph produced by all agents is $G_k = G_k^1 \cup G_k^2 \cup \ldots \cup G_k^N$. Therefore, after $k$ rounds of simulation, $G$ gradually evolves into $G'$: $G' = G_0 \cup G_1 \cup G_2 \cup \cdots \cup G_k$. Specifically, the GAG framework employs a three-step simulation workflow, as illustrated in Figure 1: **(1) Agent Formulation**: Diverse agents with unique profiles and memory capacities are initialized. **(2) Pairwise Interaction**: Agents engage in pairwise interactions through a virtual environment. **(3) Graph Extraction**: $G$ is extracted as abstract representations of interaction data. Additionally, we implement a **Parallel Acceleration** framework to enhance the efficiency of the simulation process.

## 3.2 AGENT FORMULATION

To achieve a credible simulation of human behaviors, we adhere to the general paradigm for constructing LLM-based Agents (Park et al., 2023), each agent $a_l$ is equipped with three key components: **Agent Profile**, denoted as $p_l$; **Memory Stream**, denoted as $m_l$; and **Action State**, denoted as $s_l$.

**Agent Profile** Previous research (Chan et al., 2024) has shown that persona-enhanced prompting effectively guides LLMs in generating distinctive role-play synthetic data. To develop agents with distinct roles, we create agent profiles $p_l$, which store various aspects of human personal information in textual format. Each agent $a_l$ is initialized using its corresponding profile. These profiles are derived from both real and LLM-generated datasets, with the detailed information provided in Appendix A.1. We then construct heterogeneous agents with diverse profiles, enhancing the realism and scope of our simulations.

**Memory Stream** Memory stream $m_l$ collects activity history of $a_l$ in the simulation environment. The memory component acts as a vital reference, enabling the agent to make informed decisions based on past actions and interactions within the simulation environment. We organize the memory stream using reflection (Shinn et al., 2023) and summarization techniques. This design allows the agent to adaptively learn from past interactions.

**Action State** $a_l$ is equipped with action state $s_l$ its interaction with the environment, which consists of two states: when $s_l = s_{act}$, $a_l$ engages in interaction; when $s_l = s_{inc}$, $a_l$ doesn't.

## 3.3 PAIRWISE INTERACTION

To construct the graph structure, we collect pairwise interaction data between agents and derive the graph from it. The data corpus, denoted as $C$, consists of text outputs generated by the agents, which also form the environment observed by all agents. For initialization of $C$, we first transfer seed graph $G$ into its textual form, denoted as $C$. To achieve this, we utilize an environment template $T$ to convert the graph $G(\mathcal{V}, \mathcal{E}, \mathcal{X})$ into its textual format. We map each vertex $v_i$ to a corresponding real-world entity, creating a set of textual descriptions for each entity Feng et al. (2024), details in Appendix A.2. For $v_i$, we consider its neighboring edges $E_i = \{(v_i, v_j) \mid v_i, v_j \in \mathcal{V}\}$, which compasses all edges connecting $v_i$ to its neighbors; and the vertex's text attribute $x_i$. Consequently, $C$ is represented as:

$$C = \{c_1, c_2, \ldots, c_n\} \quad \text{where} \quad c_i = T(v_i, E_i, x_i),$$

We define this graph-to-text transformation process as the function $C = f(G)$. To model the graph evolution, we continuously enrich $C$ by agent-wise interactions. Starting with the initial corpus $C_0 = f(G)$. For the $k$-th simulation round, $C$ is updated with additional textual-form graph generated by agents, denoted as $C_k$. After $k$ rounds, the corpus evolves from $C$ to $C'$: $C' = C_0 \cup C_1 \cup C_2 \cup \cdots \cup C_k$. At below, we illustrate the process of generating $C_k$. In the $k$-th simulation round, two main operations are performed: **State Planning** and **Act with S-RAG**.

**State Planning** In real-world scenarios, human activity exhibits varying levels of frequency. For instance, the user's activity frequency on online social media typically follows a Pareto distribution (Guo et al., 2009). Consequently, we first label the top 20% agents as *core*, while the remaining agents as *regular* based on action history $m_l$. This label is stored in $p_l$ (see details in Appendix A.2). $p_l$ and $m_l$ serves as input to the LLM, and each agent determines its action state $s_l = \text{LLM}(a_l \mid p_l, m_l)$.

**Act with S-RAG** To interact with other agents, $a_l$ ideally requires access to the complete $C$ and choose subset $\tilde{o}_l \subset C$ as desired environment feedback. However, since $|C|$ scales with the number of agents $N$, which can be extremely large for large-scale interaction simulations, the computational cost becomes prohibitive due to the context length limitations of LLMs. Fortunately, empirical studies show that humans hold thresholds for both information processing (Yau et al., 2020) and information spreading (Singh et al., 2013). Consequently, when agent $a_l$ decides to act, it is provided with only partial environmental information, denoted as $o_l \subset C$. Our objective is to ensure that $o_l$ determined by S-RAG closely aligns with $\tilde{o}_l$. Following traditional RAG (Cuconasu et al., 2024) framework, S-RAG is divided into three processes as shown in Algorithm 1:

**(1) Index Process:** Given the environment data corpus $C = \{c_1, c_2, \ldots, c_n\}$, where each $c_i$ is stored as a text document. we first convert this corpus into a set of embedding vectors $E$ using an embedding model $\text{encoder}(\cdot)$. The process involves transforming each $c_i$ into $d$-dimensional vectors through

---

**Algorithm 1** Process of S-RAG for the $k$-th simulation round.

---

**Require:** Interaction Data Corpus $C$, large language model LLM, Action Template $T_a$
1: **Step 1: Index Process**
2: $\mathcal{E} = \{\text{encoder}(c), c \in C\}$,
3: $C_k = \emptyset$
4: **for** $l \in [1, N]$ **do**
5:     **if** $s_l == s_{act}$ **then** $Q_l = \text{LLM}(a_l \mid m_l, s_l = s_{act})$
6:     **else** continue
7:     **end if**
8:     **Step 2: Query Process**
9:     $o_l = \emptyset$,
10:     **for** $q \in Q_l$ **do**
11:         $o_l^q = \text{RECALL}(q, C)$,
12:         $o_l^q = \text{RERANKING}(o_l^q, a_l)$,
13:         $o_l.\text{append}(o_l^q)$,
14:     **end for**
15:     **Step 3: Generation Process**
16:     $c^{kl} = \text{LLM}(a_l \mid s_l, o_l, m_l, T_a)$.
17:     $C_k = C_k \cup \{c^{kl}\}$
18: **end for**
19: $C = C \cup C_k$

---

encoder($\cdot$) and subsequently storing these vectors in a vector database Douze et al. (2024):

$$E = \{e_1, e_2, \ldots e_n\} \quad \text{where} \quad e_i = \text{encoder}(c_i) \in \mathbb{R}^d$$

This process constructs a database-based interaction environment for agents, thereby providing them with environmental information.

**(2) Query Process:** For agents in an active state, they can freely access environmental information. For agent $a_l$, obtaining the most relevant information requires first reflecting on its memory $m_l$. This reflection serves as input to the LLM, for $a_l$ to create a query set: $Q_l = \text{LLM}(a_l \mid m_l, s_l = s_{act})$. For $q \in Q_l$, we first convert $q$ into embedding vector $e_q = \text{encoder}(q)$. We retrieve the most relevant information $o_l^q$ for each $q$. This collectively forms the environment feedback $o_l := \{o_l^q, q \in Q_l\}$. The retrieving process of $o_l^q$ is divided into two stages: 1. In Recall stage, we filter out the top $N_r$ documents using the vector retriever, by measuring the embedding similarity between $C$ and $q$:

$$\text{topk}_{c \in C} \text{Sim}(q, c) \to o_l^q \quad \text{where} \quad \text{Sim}(q, c) = \frac{e_q \cdot e_c}{\|e_q\| \cdot \|e_c\|},$$

where $o_l^q$ represents the most relevant documents for $q$. We adopt cosine similarity to form the similarity function $\text{Sim}(\cdot)$. 2. In ReRanking stage, we refine and organize $o_l^q$ according to the personal preferences of agent $a_l$, which is divided into two phases: (1) Coarse Ranking: we reorder $o_l^q$ based on the interaction data from agents labeled as *core*, which are are prioritized in $o_l^q$. (2) Fine Ranking: we reorder $o_l^q$ according to the agents' personal preference information in $p_l$. For example, for $a_l$ with author role and expertise in AI, documents focused on AI-related topics are prioritized in $o_l^q$. Details are provided in Appendix A.2.

**(3) Generation Process:** Based on action state $s_l$, memory stream $m_l$, environment feedback $o_l$, the agent generates textual entity descriptions within the environment. Specifically, we instruct $a_l$ to generate coherent and logical response using LLM guided by an action template $T_a$. For the $k$-th simulation round, the output generated by $a_l$ is denoted as $c^{kl}$, which constitute a textual-form graph. Thus the process can be expressed as $c^{kl} = \text{LLM}(a_l \mid s_l, o_l, m_l, T_a)$.

Finally, $m_l$ is updated with $c^{kl}$, thereby refining $a_l$'s perception of the environment. In summary, for the $k$-th simulation round, the formulation process of $C_k$ can be mathematically formulated as: $C_k = \text{concat}(c^{k1}, c^{k2}, \ldots, c^{kN})$.

## 3.4 GRAPH EXTRACTION

Through iterative rounds of interaction simulations, we progressively enhance $C'$ using synthetic interaction data. Consequently, we can reverse mapping $C'$ into $G'$ through regex matching, which

can be formulated as $G' = g(C')$. Specifically, for each $c \in C$, we establish three mapping functions to extract the graph components: (1) corpus-to-node mapping function $g_v : C \rightarrow \mathcal{V}$. (2) corpus-to-edge mapping function $g_e : C \rightarrow \mathcal{E}$. (3) corpus-to-text attribute mapping function $g_x : C \rightarrow \mathcal{X}$. The specific mapping functions are dependent on the target graph type, as detailed in Appendix A.3. For example, in a citation network where $C$ consists of research papers, $g_v$ provides a one-to-one mapping from each paper $c \in C$ to a node; $g_e$ maps all citations within each paper to edges; $g_x$ maps all paper contents to node-wise text attributes. As a result, the subgraph $G_k$ generated in the $k$-th round can be expressed as:

$$G_k = G_k^1 \cup G_k^2 \cup \ldots \cup G_k^N,$$
$$G_k^l = (g_v(c^{kl}), g_e(c^{kl}), g_x(c^{kl})), \quad \text{where} \quad l \in [1, 2, \ldots, N].$$

### 3.5 PARALLEL ACCELERATION

The S-RAG allows us to significantly reduce the interaction rounds between agents. However, there remains technical barriers in supporting agent interaction simulation at the scale of $n = 1e^5$. Additionally, we note that the inference time of LLMs is substantial, leading to prolonged IO wait times for idle LLM-based agents. Various solutions have been proposed to address this issue, such as async mechanisms in Langchain (Kansal, 2024) and ACTOR architectures in Agentscope (Gao et al., 2024). For further acceleration, we propose a Nested ACTOR Parallel Processing Schema (N-ACTOR). As highlighted by (Clauset et al., 2004), network structures often display tightly connected communities with loosely connected inter-community links. In the context of GAG, we categorize agents into distinct groups marked by strong internal interactions and weaker intergroup interactions. Agent groups can run in parallel on different CPU cores of a computation machine with $P$ ports. The specific algorithm of N-ACTOR is detailed in the Appendix A.4.

## 4 EXPERIMENT

In network science, there has long been an interest in graph structures that emerge within scientific, technological, and sociological contexts (Leskovec et al., 2007). To assess our framework, we conduct scenario modeling of these representative domains for graph generation, each illustrating different types of human activity: (1) **Creative Activity** (CA): This includes the creation of items like academic papers. The agent acts as an author to interact with an academic paper database, producing networks such as Citation, Bibliographic Coupling, Co-Citation, Author Citation, and Co-Authorship networks (Garfield, 2000). Simulation stops at the citation network reaching 1e4 nodes. (2) **Item Interaction Activity** (IA): This includes user-item interactions, including rating movies, purchasing products and etc. The agent acts as a user interacting with an item database, producing networks such as the Movie Rating and User Projection networks (Zhou et al., 2007). Simulation stops at the movie rating network reaching 1e5 edges. (3) **Social Activity** (SA): This includes interaction and communication behavior. The agent acts as a user interacting with an online social media platform database (e.g., Twitter), producing networks such as Follow, Friend, and Action networks (De Domenico et al., 2013). The simulation stops after 5 simulation rounds. For each simulation scenario, we extract corresponding networks from each simulation scenario.

**Evaluation Protocol** To evaluate the effectiveness of the GAG, we assess three key aspects: First, we compare the generated graph structures with real-world networks at both macro and micro scales to verify the similarity between the generated and real-world graph evolution. Next, we evaluate the effectiveness of text features for the generated graph using a node classification task. Finally, we assess the scalability of GAG in terms of graph size and generation efficiency. Details on the evaluation hyperparameters and metrics are provided in Appendix B.1.

### 4.1 EVALUATION OF STRUCTURE ALIGNMENT

Unlike traditional and deep learning-based graph generation methods, GAG does not rely on pre-defined rules or structural distributions from training datasets. Instead, graph structures naturally emerge from interactions between agents, making the evaluation of these generated structures crucial for assessing interaction rationality. In this paper, we investigate the generated graph structures from both macro and micro perspectives: At the macro level, we examine the graph structure dynamics in

the graph evolution, and align our observations with established network science theories. At the micro level, we use existing graph generation models as baselines to evaluate the effectiveness of GAG in capturing micro structural metrics of graph evolution.

**Macro-Level Evaluation** For macro-level structural characteristic alignment, we examine three structural characteristic observed in real-world networks (Albert & Barabási, 2002): Power-law Distribution, Small-world Phenomenon and Shrinking Diameter. Four additional structural characteristics are detailed in Appendix B.2.

**(1) Power-law Distribution:** The degree distribution of scale-free networks often follows *power-law distribution* (Barabási & Albert, 1999), which is commonly observed in citation networks and social networks. In simulations with GAG, the generated citation, author-citation, and action networks also exhibit this characteristic. We adhere to the established criteria for evaluating whether the graph's degree distribution follows a power law: $D_k < 0.1$ (Alstott et al., 2014). As shown in Figure 2, the degree distribution of these networks follow a *Power-law* distribution with exponent parameter $\alpha \in [1.90, 2.16]$, closely aligned with $\alpha \in [2, 3]$ from empirical studies (Clauset et al., 2009).

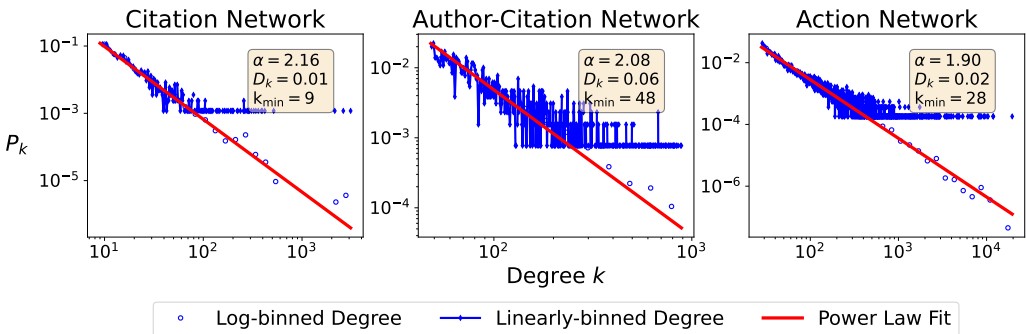

Figure 2: The *Power-law* Distribution of degrees in scale-free networks. The degree $k$ is plotted against the probability density function $p_k$ on a log-log scale. $\alpha$ denotes the exponent parameter, $k_{min}$ represents the minimum cut-off $k$ (Alstott et al., 2014).

**(2) Small World Phenomenon:**

Table 1: $\bar{c}c$ of the generated networks, and the ratio to Erdös-Rényi and Barabási-Albert graph model. A dash (—) signifies that $\bar{c}c = 0$ for the graph model.

| | Ratio to Random Graphs | | | | |
|---|---|---|---|---|---|
| | Citation | Bib-Coupling | Co-Citation | Author Citation | Co-Authorship |
| Erdös-Rényi | 301.08 | 7.46 | 275.97 | 39.82 | 234.81 |
| Barabási-Albert | — | 4.40 | 44.87 | 11.19 | 17.59 |
| | Action | Follow | Friend | Movie Rating | User Projection |
| Erdös-Rényi | 784.93 | 3961.83 | 19768.58 | 0.00 | 5.78 |
| Barabási-Albert | 73.97 | 443.80 | 1391.47 | 0.00 | 2.82 |

Real-world networks exhibit a *small world* phenomenon (Mislove et al., 2007; Watts & Strogatz, 1998), characterized by a small diameter and a high clustering coefficient. Based on the structure of the generated graphs, we construct two types of random graphs with consistent average degree: Erdős-Rényi (ER) (Erdos et al., 1960) and Barabási-Albert (BA) graphs (Barabási & Albert, 1999). Table 1 compares the average clustering coefficient ($\bar{c}c$) of the generated graphs with that of the random graphs. The results indicate that generated social networks (i.e., follow, friend, and action networks), exhibit a significantly higher $\bar{c}c$ than those of the random graphs. Additionally, as shown in Table 4, the generated networks exhibit a small diameter, $D_e \in [1.17, 11.66]$, consistent with the *six degrees of separation phenomenon* observed in real-world networks (Leskovec et al., 2007; Broder et al., 2000). The high $\bar{c}c$, combined with small $D_e$, confirms these networks exhibit small-world characteristics.

**(3) Shrinking Diameter:** The *shrinking diameter* is a notable phenomenon in social networks (Leskovec et al., 2007), with $D_e$ decreases as the network evolves over time. We construct SA with $N = 7000$, and investigate the graph evolution processes of follow, friend, and action networks for 30 simulation rounds. We calculate the $D_e$ metric for both the generated graphs and real-world network datasets: CAIDA[1]. As shown in Figure 3a, $D_e$ decreases at a slow pace, identical to the trend observed in (Leskovec et al., 2007) and CAIDA. To explain *shrinking diameter*, models such as the Forest Fire model (Leskovec et al., 2007) utilize a modified preferential attachment mechanism, known as community-guided attachment. In GAG, the ReRanking process enhances personalized recommendations. To assess its impact on graph structure, we conduct an ablation experiment. As shown in Figure 3b, we observe an initial increase in $D_e$ followed by a rapid decline within friend networks. Notably, upon removing the ReRanking, the $D_e$ trends upwards from 2.6 to 2.96, indicating that ReRanking fosters community-guided attachment and shrinking diameter.

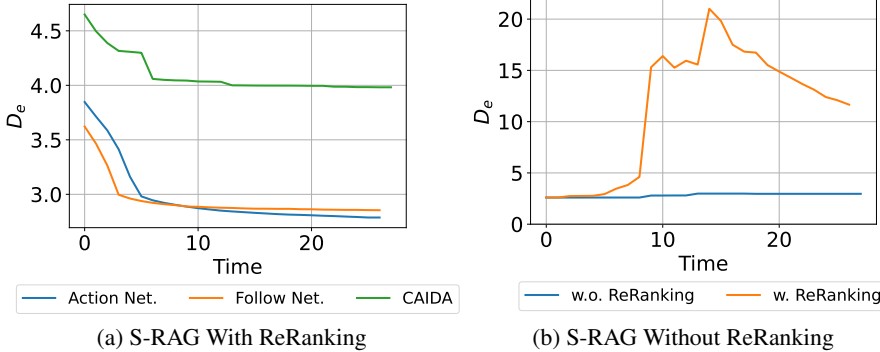

(a) S-RAG With ReRanking     (b) S-RAG Without ReRanking

Figure 3: The *Shrinking Diameter* phenomenon simulated by GAG; The left figure demonstrates that as the graph evolves, $D_e$ gradually decreases in action and follows networks; The right figure presents an ablation experiment of the ReRanking, demonstrating its effect on $D_e$ in friend network.

**Micro-Level Evaluation** [Revision: For micro-level structural characteristic alignment, we examine the structural characteristics of generated graphs. For baselines, we select the efficient graph generation model for large-scale graph generation. Specifically, we partition the Citeseer network (Sen et al., 2008) into $G_{<t}$ and $G_{>t}$ based on time $t$. We then expand the small subgraphs of $G_{<t}$ and compare the generated graphs to the large subgraphs of $G_{>t}$. Details in Appendix B.3.

Table 2: Comparison with existing graph generation models for graph expansion task. For GRAN and GraphMaker, the generated graphs fail to converge to a power-law distribution.

|  | MMD.D↓ | MMD.C↓ | MMD.S↓ | MMD.O↓ | $D_k$ | $\alpha$ | Valid↑ | GEM |
|---|---|---|---|---|---|---|---|---|
| CiteSeer | - | - | - | - | $0.06_{\pm 0.0}$ | $2.38_{\pm 0.0}$ | 1.0 | - |
| Erdös-Rényi | 0.26 | 1.41 | 0.56 | 1.41 | $0.1_{\pm 0.01}$ | $3.72_{\pm 0.13}$ | 0.0 | 0.34 |
| Barabási-Albert | 0.20 | 1.41 | 0.26 | **1.02** | $0.04_{\pm 0.01}$ | $2.38_{\pm 0.04}$ | **1.0** | 0.42 |
| Small-World | 0.72 | 1.36 | 0.59 | 1.41 | $0.42_{\pm 0.01}$ | $2.03_{\pm 0.0}$ | 0.0 | 0.32 |
| BiGG | 0.63 | 1.13 | 0.65 | 1.23 | $0.27_{\pm 0.01}$ | $1.69_{\pm 0.01}$ | 0.0 | 0.33 |
| GRAN | 0.36 | 0.55 | 0.72 | 1.41 | - | $4.16_{\pm 0.39}$ | 0.0 | 0.36 |
| BwR | 0.49 | 1.41 | 0.66 | 1.41 | $0.07_{\pm 0.09}$ | $4.46_{\pm 0.02}$ | 0.0 | 0.32 |
| GraphMaker | 0.47 | 1.41 | 0.83 | 1.41 | - | - | 0.0 | 0.22 |
| L-PPGN | 0.76 | 1.19 | 0.78 | 1.05 | $0.39_{\pm 0.03}$ | $1.36_{\pm 0.02}$ | 0.0 | 0.33 |
| GAG | **0.16** | **0.19** | **0.32** | **1.02** | $0.08_{\pm 0.01}$ | $2.37_{\pm 0.03}$ | **1.0** | **0.47** |

We define the Valid metric to measure the proportion of valid power-law expanded graphs and the GEM metric to evaluate the overall effectiveness of structure alignment in expanded graphs, details in Appendix B.3 As shown in Table 2, the expanded graphs by GAG adhere to a power law distribution, with $\alpha = 2.39$ close to the seed graph. In contrast, deep-learning-based models tend to overfit the training graph and fail to generate graphs that follow the power-law distribution effectively. Surprisingly, GAG outperforms most baseline models in MMD metrics, which is notable given that no explicit graph structure constraints are provided to the agents. Regarding GEM, GAG surpasses

---

[1]https://sparse.tamu.edu/SNAP/as-caida

the best performing baseline by 11%. This suggests that GAG effectively simulates human behavior patterns, generating graphs that closely resemble the structural characteristics of real-world networks. ]

## 4.2 EVALUATION OF TEXTUAL-ATTRIBUTE ALIGNMENT

[Revision: GAG generates text-rich dynamic graphs by collecting interaction data among various agents, simulating the process of gathering textual features for real-world networks. To evaluate whether the generated graphs preserve the text-structure correlations of the seed graph, we adopt Graph Neural Networks (GNNs) benchmarking tasks of node classification Yoon et al. (2023). Specifically, we employ different Graph Neural Networks (GNNs) to train on both $G'$ and $G$ for the node classification task, measuring the accuracy gap between the two. We denote the accuracy of the GNN on $G$ as $Acc_b$ and the accuracy on $G'$ as $Acc$. The accuracy gap is calculated as $\Delta Acc = |Acc - Acc_b|$, which measures the fidelity of GAG in preserving the text-structure of $G$; a smaller $\Delta Acc$ indicates better preservation. For benchmarking, we select four representative GNN architectures and construct graphs with eight distinct relationships between graph structures and textual features. Details of the experimental setup are provided in Appendix C.

Table 3: Benchmarking different graph generation models on node classification task.

| | $\Delta ACC \downarrow$ | | | |
|---|---|---|---|---|
| | GAT | GCN | GCN2 | GraphSage |
| SF.random | $13.4_{\pm 3.52}$ | $15.04_{\pm 1.99}$ | $9.58_{\pm 0.88}$ | $7.0_{\pm 0.92}$ |
| F.random | $24.11_{\pm 2.84}$ | $23.2_{\pm 2.25}$ | $9.33_{\pm 1.36}$ | $7.13_{\pm 1.73}$ |
| S.random | $18.85_{\pm 2.55}$ | $21.77_{\pm 2.04}$ | $2.24_{\pm 1.58}$ | $3.15_{\pm 1.84}$ |
| BiGG+LLM | $39.39_{\pm 4.51}$ | $34.06_{\pm 6.62}$ | $4.73_{\pm 4.43}$ | $3.38_{\pm 3.54}$ |
| GRAN+LLM | $5.31_{\pm 3.63}$ | $6.08_{\pm 5.12}$ | $3.46_{\pm 3.39}$ | $4.35_{\pm 2.47}$ |
| BwR+LLM | $35.28_{\pm 2.91}$ | $38.52_{\pm 4.68}$ | $4.83_{\pm 4.47}$ | $7.59_{\pm 4.30}$ |
| GraphMaker+LLM | $4.04_{\pm 3.49}$ | $\mathbf{2.83}_{\pm 4.02}$ | $3.59_{\pm 4.24}$ | $4.17_{\pm 4.03}$ |
| L-PPGN+LLM | $38.39_{\pm 6.39}$ | $32.64_{\pm 3.46}$ | $3.96_{\pm 3.62}$ | $3.85_{\pm 3.75}$ |
| GAG | $\mathbf{2.26}_{\pm 1.19}$ | $3.61_{\pm 1.32}$ | $\mathbf{0.49}_{\pm 1.50}$ | $\mathbf{0.09}_{\pm 1.74}$ |

As shown in Table 3, the random baselines perform the worst, highlighting the importance of the tight coupling between graph text attributes and graph structure. Existing models such as GRAN and GraphMaker demonstrate strong performance in GNN benchmarking tasks, consistent with the findings in Li et al. (2024a). However, GAG consistently outperforms these baselines, achieving an average improvement of 1.45 in $\Delta Acc$ across GNNs, with $\Delta Acc$ values ranging from 0.09 to 3.61. These results demonstrate GAG's effectiveness in capturing the intricate text-structure correlations. ]

## 4.3 SCALABILITY OF GAG

To evaluate the scalability of the framework, we assess GAG in terms of both graph generation scale and efficiency.

**Graph Generation Scale** In addition to expanding seed graphs, GAG can generate graphs from scratch using LLM-generated agent profiles, eliminating the need for external data collection. We employ this method to generate social networks and produce networks with nearly 100,000 nodes while expanding bibliographic coupling networks to 12.2 million edges. While existing models typically limit graph generation to a maximum of 5,000 nodes (Bergmeister et al., 2024; Liao et al., 2019), specialized models designed for sparse large graphs can only generate simple grid-structure graphs with up to 100,000 nodes (Dai et al., 2020). This highlights the capability of GAG to support large-scale agent interaction simulation so as to model large-scale graph evolution. [2] As shown in Table 4, we calculate the structural metrics of all generated networks. Similar to the assortative-

---

[2]To intuitively illustrate the graph evolution process modeled by GAG, we choose the SA simulation scenario and visualization in `https://anonymous.4open.science/r/GraphAgent-2206/visualization/social_network.mp4`

Table 4: The structural metrics for graphs generated by GAG.

|  | Citation | Bib-Coupling | Co-Citation | Author Citation | Co-Authorship |
|---|---|---|---|---|---|
| $|\mathcal{V}|$ | 1.14e+04 | 1.09e+04 | 3.93e+03 | 5.01e+03 | 5.01e+03 |
| $|\mathcal{E}|$ | 3.63e+04 | 1.22e+07 | 3.27e+04 | 2.41e+05 | 2.08e+04 |
| $\bar{c}c$ | 0.08 | 0.77 | 0.59 | 0.38 | 0.20 |
| $r$ | -0.10 | 0.09 | -0.10 | -0.18 | 0.32 |
| $D_e$ | 5.19 | 2.94 | 3.89 | 3.44 | 5.77 |
|  | Action | Follow | Friend | Movie Rating | User Projection |
| $|\mathcal{V}|$ | 9.97e+04 | 9.96e+04 | 9.96e+04 | 4.17e+03 | 3.91e+03 |
| $|\mathcal{E}|$ | 9.07e+05 | 1.53e+06 | 5.01e+05 | 3.25e+04 | 9.04e+05 |
| $\bar{c}c$ | 0.07 | 0.61 | 1.00 | 0.00 | 0.34 |
| $r$ | -0.03 | 0.06 | 0.59 | -0.54 | -0.11 |
| $D_e$ | 2.79 | 2.85 | 11.66 | 2.98 | 1.17 |

mixing patterns discovered in real-world networks (Newman, 2002), the citation network exhibits negative assortativity, whereas the co-authorship network displays positive assortativity.

**Graph Generation Efficiency** We evaluate efficiency by measuring the impact of the number of agents ($N$) and the number of ports ($P$) on the runtime of the framework, specifically the time required to generate a single interaction. Tests are conducted on a machine with 96 CPU cores and 376GB of memory. As shown in Table 5, when $P$ is held constant, the time to generate one interaction data decreases as $N$ increases. The most significant time reduction observed in the CA simulation, where agents are grouped by paper authorship, which maximizes the efficiency of the N-ACTOR component. Conversely, as shown in Table 6, when $N$ remains constant, the time to generate a single $c$ decreases as $P$ increases. Notably, the time decreases by 90.4% when $P > 1$ compared to $P = 1$ at least, demonstrating the acceleration effectiveness of N-ACTOR component.

Table 5: The time cost (*min*) of agents for generating one interaction data with $P = 24$.

| N | CA | IA | SA |
|---|---|---|---|
| 5 | 0.2700 | 0.0150 | 0.0120 |
| 10 | 0.2300 | 0.0112 | 0.0119 |
| 20 | 0.1700 | 0.0052 | 0.0119 |
| 40 | 0.0910 | 0.0054 | 0.0060 |
| 5→40 | ↓**66.3%** | ↓**64.0%** | ↓**50.0%** |

Table 6: The time cost (*min*) of 40 agents for generating one interaction data with different $P$.

| P | CA | IA | SA |
|---|---|---|---|
| 1 | 3.6250 | 0.0683 | 0.0623 |
| 4 | 0.1470 | 0.0068 | 0.0112 |
| 16 | 0.1160 | 0.0053 | 0.0109 |
| 24 | 0.0910 | 0.0054 | 0.0060 |
| 1→24 | ↓**97.5%** | ↓**92.1%** | ↓**90.4%** |

## 5 Conclusion

In this study, we present GAG, a novel and general framework designed for generating large-scale graphs with human behavior simulation. Our comprehensive experiments show that GAG can generate large-scale graphs with high quality and efficiency, with specified focus on graphs arise from scientific, technological and sociological context. The generated graphs exhibit seven macro-level characteristics of real-world networks, including power law, small world and shrinking diameter. In the citation network expansion task, GAG demonstrates superior performance compared to existing deep-learning-based graph generation methods in accurately capturing the power-law distribution property. Furthermore, we present the S-RAG algorithm for simulating diverse human interaction processes at scale, complemented by the N-ACTOR for parallel acceleration, achieving a speed-up of at least 90.4%. Our framework successfully produces high-quality graphs with up to nearly 100,000 nodes or 10 million edges. Overall, GAG serves as a promising step towards efficient large-scale human interaction simulation and graph generation.

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

# A  DETAILS OF GAG

## A.1  AGENT FORMULATION

To demonstrate the versatility of the GAG Framework, we build three graph generation tasks for different human activities in our experiments, the concrete settings are as follows: (1) CA: In this scenario, agents act as authors interacting with a paper database and generate the following networks: Citation, Bibliographic Coupling (Bib-Coupling), Co-Citation, Author Citation, and Co-Authorship networks (Garfield, 2000). (2) IA: In this scenario, agents act as reviewers interacting with both online movie databases and offline movie databases and generate the following networks: the Movie Rating and the User Projection networks (Zhou et al., 2007). (3) SA: In this scenario, agents act as users interacting with a Twitter-like online social media database and generate the following networks: Follow, Friend, and Action networks (De Domenico et al., 2013).

For agent construction, we use both real and LLM-craft data for agent profiles. Since the original datasets lack certain node attributes (e.g., Citeseer is missing author information and article content), we crawl for the necessary node attributes to enrich text attributes for Citeseer Sen et al. (2008). In addition, we use handcraft instruction and COT (Wei et al., 2022) to guild LLM in generating synthetic agent profiles.

For various simulation scenarios, we develop agents assigned distinct roles, including paper authors, movie watchers, and Twitter users. These agents interact with an environment modeled on historical interaction data. Specific details of the settings are provided in Table 7.

For the LLM backbone, we have chosen the open-source model of Llama-3-70BAI@Meta (2024) for the large-scale graph generation in macro-level structure alignment and textual-attribute alignment experiments. For micro-level structure alignment, we select the closed-source model of GPT-3.5-turbo for a more accurate simulation of human behaviors. Additionally, we select Reimers & Gurevych (2019) [3] as the encoder in S-RAG.

Table 7: The Agent Settings and Interaction Data Types for Different Simulation Scenarios.

| Scenario | Agent Profile | Agent Role | Environment Data | Action Type |
|---|---|---|---|---|
| CA | Citeseer (Sen et al., 2008), Cora (Sen et al., 2008) LLM-Generated | Paper Author | Papers | Paper Writing |
| IA | Movielens (Harper et al., 2016), LLM-Generated | Movie Watcher | Movies | Movie Rating |
| SA | LLM-Generated | Twitter User | Tweets | Tweet Sending |

## A.2  PAIRWISE INTERACTION

**State Planning**  In online social media, there is a significant difference in the influence of content shared by core users versus general users (Mislove et al., 2007; Mou et al., 2024). According to the Pareto distribution, core users should constitute approximately 20% of the total users in the current social network. We denote the agents labeled as *core* as $HUB$, as a result, the proportion of these agents is $|HUB|/|V|$. To investigate this, we adjust the proportion of agents labeled as *core*. Details in Appendix E.2.

**Act with S-RAG**  We define the environment template $T$ to transform each node in text-attributed graph into a natural language representation. For instance, in a citation network, graph nodes represent academic papers, whereas in a movie-rating network, they represent movies. This graph-to-text transformation process leverages the node features and optionally considers edge features to enhance the representation. To better illustrate this process, we present the environment template $T$ for various simulation scenarios below:

---

[3]https://huggingface.co/sentence-transformers/all-MiniLM-L6-v2

Table 8: The environment prompt template for *CA*

**Node Feature** $(v_i, x_i)$: Academic paper.
Title: <Title>
Topic: <Topic>
Abstract: <Abstract>
Author: <Author>.
**Edge Feature** $(E_i)$: The citation papers of paper $v_i$.

Table 9: The environment prompt template of movies for *RA*

**Node Feature** $(v_i, x_i)$: Movie.
Title: <Title>
Genres: <Genres>
Content: <Movie Abstract>
Movie Rating: <Movie Rating>
**Edge Feature** $(E_i)$: NULL.

Table 10: The environment prompt template of movies for *SA*

**Node Feature** $(v_i, x_i)$: Tweets.
Tweet ID: <Tweet ID>
User: <Tweet User>
Tweet: <Tweet Content>
**Edge Feature** $(E_i)$: NULL.

To identify the most relevant information to $a_l$, we propose the S-RAG algorithm. For detailed explanation of the query process in S-RAG:

In the recall stage, we initially retrieve $o_l^q$ as the candidate documents for $a_l$, which serves as the initial environmental feedback. This step we filter out $N_r$ candidate documents.

In order to align $o_l^q$ with agent's personal preference more accurately, we adopt the reranking stage for post-processing of $o_l^q$, which is divided into two phases: (1) Coarse Ranking: We reorder $o_l^q$ based on whether the interaction data was generated by agents labeled as *core*. Candidate documents originating from *core* agents are positioned at the forefront of $o_l^q$, while those from non-core agents are placed towards the end. (2) Fine Ranking: We further reorganize $o_l^q$ based on the personal preferences of the agents, as represented by $p_l$. For CA simulation, the filter items include topics of the academic papers; for RA simulation, the filter items include movie genres; and for SA simulation, the filter items include attributes of posted tweets, such as friends, topics, and follows. Ablation study on the filter items is detailed in Appendix. E.2.

Assuming we add $k$ agents to the GAG framework per round and aim to ultimately include $N$ agents, it would require $\frac{N}{k}$ simulation rounds. In the $j$-th round, the number of agents within GAG would be $N' = k \times j$. Without S-RAG, if we assume each of these $N'$ agents needs to interact pairwise, the interaction number for one agent to interact with all others is $O(N')$. Thus, the total interaction number for $N'$ agents interacting pairwise is $O(N'^2)$. Therefore, the total pairwise interaction number can be expressed as:

$$O(k^2 + (2k)^2 + \ldots + \left(\frac{N}{k}k\right)^2) = O(N^3)$$

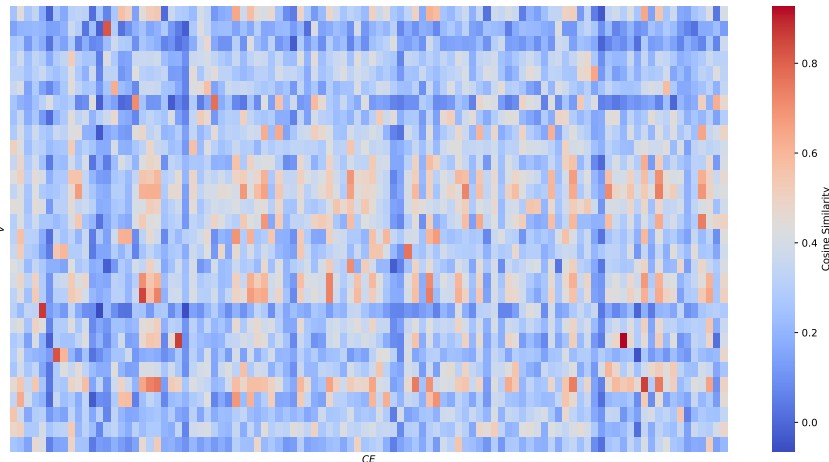

Figure 4: The cosine similarity between the embedding vectors of $E$ and $V$, where $V$ represents the embedding vectors of queries from different agents and $E$ represents the embedding vectors of texts within the database.

As illustrated in Fig.4, the agents' queries are only relevant to a small subset of $C$, i.e. each agent interacts with only a limited number of other agents. S-RAG filters out a subset $\tilde{C} \subseteq C$ of agents, where $|\tilde{C}| = N_r$. Hence, the interaction number of one agent is reduced to $O(N_r)$, for $N'$ agents to interact pairwise is $O(N' \times N_r)$. Consequently, the general interaction number in the whole simulation process is:

$$O(k \times N_r + 2k \times N_r + \ldots + \frac{N}{k}k \times N_r) = O(N^2)$$

Thus, the incorporation of S-RAG significantly reduces the pairwise interaction number from $O(N^3)$ to $O(N^2)$ during the complete simulation process.

## A.3 GRAPH EXTRACTION

In each simulation scenario, agents are exposed to different types of environmental information, which subsequently influences their activities. Prompted by $T_a$, agents are instruct to generate textual representations of graphs.

By employing reverse regular matching, we can extract the corresponding node and edge information. Depending on the specific simulation scenario, the Action Template $T_a$ are defined as follows:

Table 11: The process of *Paper Writing* for $a_l$ in CA

**Action Template ($T_a$)**
**Agent Profile:** <Agent Profile>
**Agent Memory:** <Agent Memory>
**Environment Observation**: Partial Environmental Feedback provided by S-RAG, denoted as $o_l$. Abstract information for searched papers.
**Human Instruction**: A papar should include the following attributes:
title: The title should be concise yet descriptive, providing a clear indication of the paper's topic and scope. This can be different from your topic, It is relatively accurate and clear.
keywords: These are specific terms or phrases that encapsulate the core topics of your paper. Keywords make your paper searchable within academic databases and help readers quickly understand the paper's focus areas.
abstract: The abstract is a brief summary of your research paper. It should provide an overview of the research question, methodology, results, and conclusions.
citations: A list of the paper names you want to cite.
. . .
Now write a version of your paper and cite the papers you need to cite.

---

**Respond Example**
**Node Features**: Academic paper. Title, Keywords, Abstract.
**Edge Features**: Citation papers.

Table 12: The process of *Movie Rating* in IA

**Action Template ($T_a$)**
**Agent Profile:** <Agent Profile>
**Agent Memory:** <Agent Memory>
**Environment Observation**: Partial Environmental Feedback provided by S-RAG, denoted as $o_l$. Abstract information for searched movies.
**Human Instruction**: You should give your rating scores to the movies . . .

---

**Respond Example**
**Node Features**: NULL.
**Edge Features**: Movie ratings.

In various simulation scenarios, we progressively enrich the set $C$ and subsequently extract different graph structures from it. This results in the mapping functions $g_v : C \to \mathcal{V}$, $g_e : C \to \mathcal{E}$, and $g_x : C \to \mathcal{X}$.

In the context of CA, following action template $T_a$ defined in Table 11, each time the agent generates a paper. The LLM-based agents act as authors and $C$ stores the papers. To fold graphs from the pair-wise interaction process, we define the following mapping functions:

Table 13: The process of *Tweet Sending* in *SA*

---

**Action Template** ($T_a$)
**Agent Profile:** <Agent Profile>
**Agent Memory:** <Agent Memory>
**Environment Observation**: Partial Environmental Feedback provided by S-RAG, denoted as $o_l$. Abstract information for searched tweets.
**Human Instruction**:
You can perform [Retweet/Reply/Tweet] action on these tweets. Additionally, you can follow the bloggers of these tweets:
Retweet: Retweet the tweet
Reply: Reply to the tweet
Tweet: Send a tweet
. . .

---

**Respond Example**
**Node Features**: Tweets. Tweet contents. Tweet Topics.
**Edge Features**: Actions to other tweets (retweet, reply, follow).

---

1. **Citation Network:** Let $V$ represent the set of papers, $E$ denote the citation relationships between papers, and $X$ signify the attributes of each paper.

2. **Bib Coupling:** Let $V$ represent the set of papers, $E$ represents the relationships where two papers cite the same reference, and $X$ encompasses the attributes of the papers.

3. **Co-citation:** Let $V$ represent the set of papers, $E$ indicates the relationships where two papers are cited by the same paper, and $X$ includes the attributes of the respective papers.

4. **Author Citation:** Let $V$ represent the set of authors, $E$ depicts the citation relationships among authors, and $X$ refers to the attributes of each author.

5. **Co-Authorship:** Let $V$ represent the set of authors, $E$ illustrates the collaborative relationships between authors, and $X$ characterizes the attributes of each author.

In the context of RA, following action template $T_a$ defined in Table 12, each time the agent generates a movie rating. The LLM-based agents act as movie watchers and $C$ stores the movies. To fold graphs from the pair-wise interaction process, we define the following mapping functions:

1. **Movie Rating**: Let $V$ represent the watchers and the movies, $E$ denote the movie ratings. For movie watchers, $X$ correspond to the attributes of movie watchers; for movies, $X$ correspond to the attributes of movies.

2. **User Projection**: In this setup, $V$ consists of movie watchers, $E$ refers to the number of movies rated jointly by two movie watchers, and $X$ encompasses the attributes of the movie watchers.

In the context of RA, following action template $T_a$ defined in Table 13, each time the agent generates a tweet. The LLM-based agents act as tweet users and $C$ stores the tweets. To fold graphs from the pair-wise interaction process, we define the following mapping functions:

1. **Action:** Let $V$ represent users, $E$ denote the edges indicating tweets exchanged between two users, and $X$ represent user attributes.

2. **Follow:** Let $V$ represent users, $E$ denote the edges indicating tweets that establish a follow relationship between two users, and $X$ represent user attributes.

3. **Friend:** Let $V$ represent users, $E$ denote the edges indicating tweets that establish a friend relationship between two users, and $X$ represent user attributes.

A.4   Parallel Acceleration

In order to further speed up the simulation process of GAG, we propose a Nested ACTOR Parallel Processing Schema for Agents (N-ACTOR) based on traditional ACTOR architecture (Hewitt et al., 1973). (Clauset et al., 2004) highlights that network structure often exhibits tightly connected communities with loose inter-community connections, so we decompose the agents into different groups and run them in parallel.

---

**Algorithm 2** Nested ACTOR Parallel Processing Schema for Agents

---

**Require:** $a_l \in \{a_1, a_2, \ldots, a_N\}$, Batch size $B$
  1: **Initialization:**
  2: $g \leftarrow a_l$,
  3: INITIALIZE message queues $M^g$ for $g$,
  4: Number of batches $K = \lceil \frac{N}{B} \rceil$,
  5: $s_k \leftarrow \bigcup_{b=1}^{B} g_{kb}$    for $k \in \{1, \ldots, K\}$,
  6: INITIALIZE message queues $M_k^s$ for $s_k$,
  7: **Message Handling:**
  8: **for** each supervisor actor $s$ in parallel **do**
  9:      **for** each general actor $g_{kb}$ under $s_k$ in parallel **do**
10:         Process messages in $M^g$,
11:         Update state of $g_{kb}$,
12:         Send messages to other general actors $g_{kb}$ via $M^g$,
13:      **end for**
14:      Process messages in $M^s$,
15:      Update state of $s$,
16:      Send messages to other supervisor actor via $M^s$,
17: **end for**

---

**N-ACTOR Architecture**    The objective of N-ACTOR is to divide $N$ agents into $\frac{N}{B}$ batches of size $B$, allowing for parallel execution between batches. As illustrated in Algorithm 2, the process includes two steps: **(1) Initialization**: Following ACTOR, we define actor as the instantiated encapsulation of an agent. To facilitate batch parallelization of agents, we introduce two types of actors: general actor, denoted as $g \in G$, and supervisor actor, denoted as $s \in S$. Different actors leverage different CPU cores of the computational machine. Let $P$ denote the number of ports. Each supervisor actor $s$ manages $B$ number of $g$, thus $|S| = \frac{N}{B}$. **(2) Message Handling**: Traditional ACTOR employs a message queue for agents' execution of message processing, state updating, and message sending in parallel. (Clauset et al., 2004) highlights that network structure often exhibits tightly connected communities with loose inter-community connections, N-ACTOR introduces two levels of parallel message processing: Within each batch, $s$ can perform parallel message processing through $M^s$; Between different batches, $g$ first aggregates messages for all $g$ within the batch and performs inter-batch parallel message processing through $M^g$. As shown in Figure 5, $g$ only needs to wait for the IO time of $B$ general actors instead of $N$.

## B   Evaluation of Structure Alignment

### B.1   Graph Structure Metrics

To measure the structural characteristics of graph, we use the following structural metrics:

**(1)** $|\mathcal{V}|$: measures the node number of graph $G$.

**(2)** $|\mathcal{E}|$: measures the edge number of graph $G$.

**(3)** $\bar{c}c$: average clustering coefficient, quantifies the degree to which nodes in a graph tend to cluster together [4].

---

[4] https://en.wikipedia.org/wiki/Clustering_coefficient.

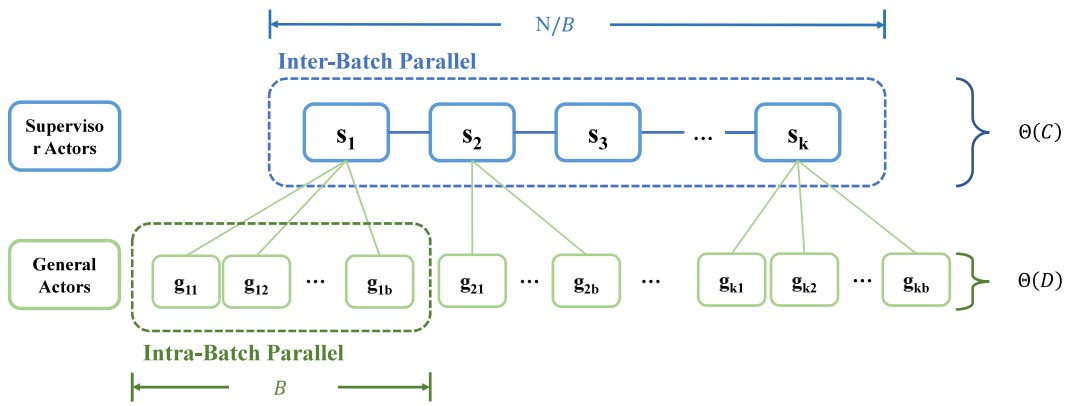

Figure 5: N-ACTOR Network Topology

**(4)** $r$: assortativity, measures the similarity of connections in the graph concerning the node degree. [5]

**(5)** $D_e$: effective diameter, defined as the minimum number of hops in which a certain percentage (typically 90% or 95%) of all connected node pairs can be reached.

## B.2 MACRO-LEVEL EVALUATION

**Periodic Variation of Degree**   In the simulation scenario of IA, we filter the review data from the Movielens-25M dataset based on the movies listed in the Movielens-1M dataset. We select the top 10 ratings for each user and discover a noteworthy phenomenon: the number of reviews in the rating network exhibits periodic fluctuations over time. By scraping the release dates of the movies and plotting their release frequency, we observe that the periodicity in the release frequency is consistent with the fluctuations in the number of reviews. To quantify the periodicity, we selected the signal-to-noise ratio (SNR) (Johnson, 2006) as our metric, considering an SNR greater than 10 dB to indicate strong periodicity and reliability. Furthermore, we observe that the degree of the generated rating graph also exhibits periodic variations consistent with the release dates of the movies. As illustrated in Figure 6, the SNR of the degree of the rating graph is 12.79 dB, surpassing the 10 dB threshold, thus demonstrating significant periodic fluctuations.

**Emergent of GCC**   In online social networks, nodes with higher degrees grow larger over time and eventually manifest a giant connected component (GCC) (Mislove et al., 2007). As illustrated in Figure 7, the proportion of the largest connected component rows steadily over time, indicating the emergence of a giant community within the social network. The network is generated by with 7000 agents.

**Friendship Paradox**   An interesting and somewhat counterintuitive phenomenon in real-world social networks is that everyone you follow or who follows you tends to have more friends and followers than you do. This phenomenon has been observed in both Twitter (Hodas et al., 2013) and the social network of Facebook (Ugander et al., 2011), applying to more than 98% of the nodes. As shown in Figure 8, the friendship paradox is most evident in the friend network, with over 90% of the nodes lying above the $y = x$ line, indicating that most users have fewer friends than their friends do. The network is generated by 5 rounds of simulation with 1e5 agents.

---

[5]https://en.wikipedia.org/wiki/Assortativity

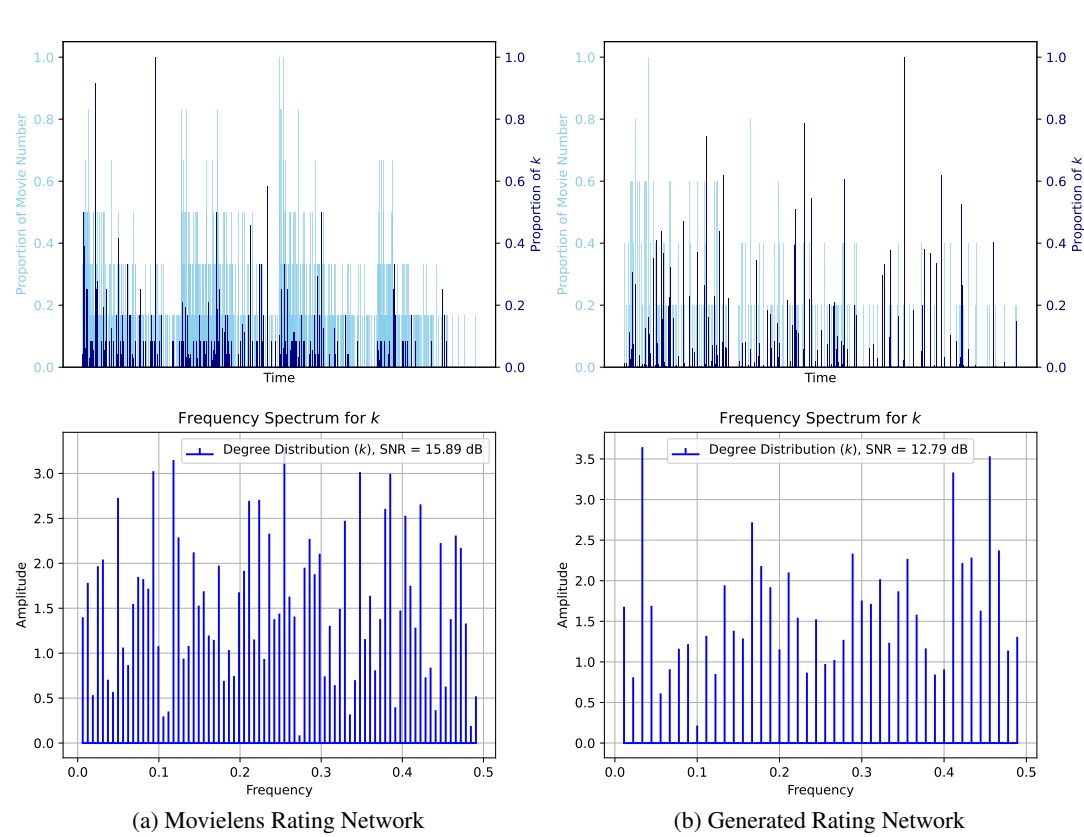

(a) Movielens Rating Network        (b) Generated Rating Network

Figure 6: Periodic Variation of Degree in Movie Rating Network; Figure 6a shows the number of released movies over time and the degree of the movie rating network over time in MovieLens dataset; Figure 6b also shows the number of released movies and the degree of the movie rating network over time in GAG.

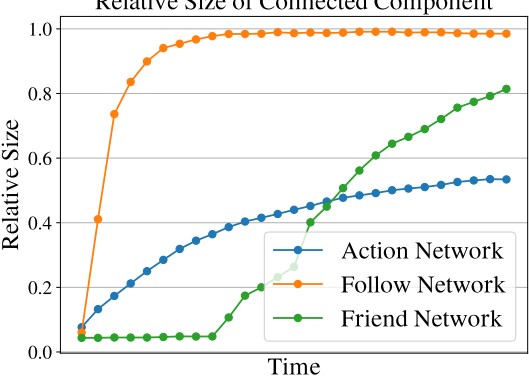

Figure 7: The proportion of the largest connected component grows steadily over time.

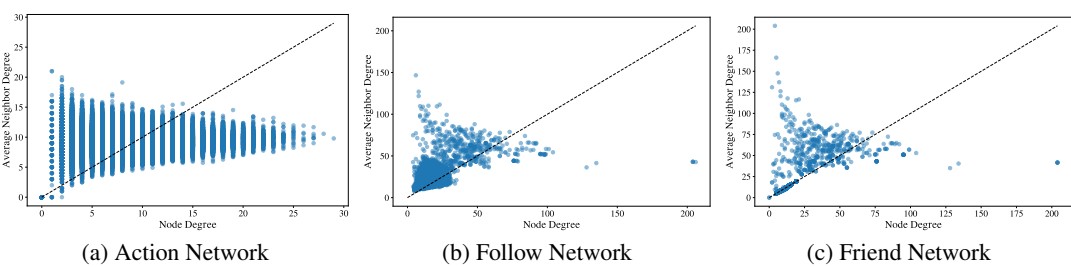

(a) Action Network      (b) Follow Network      (c) Friend Network

Figure 8: The *Friendship Paradox* phenomenon in online social networks; The figures show the average degree of node neighbors v.s. the average degree of node itself in social networks.

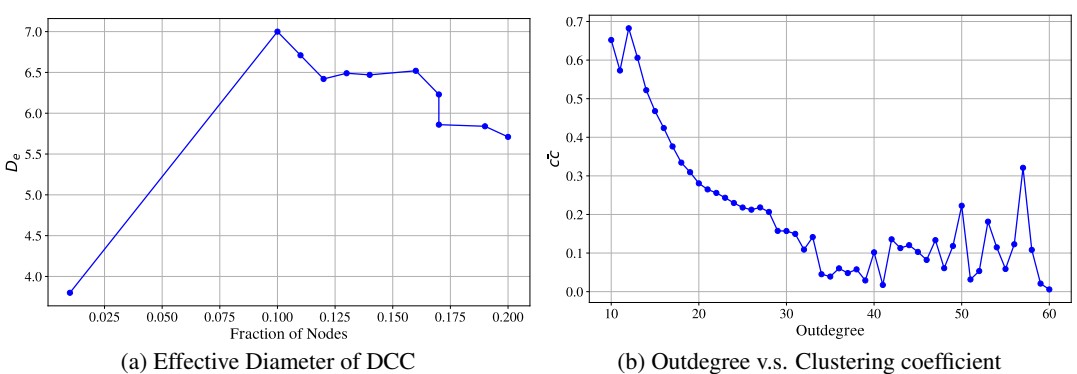

(a) Effective Diameter of DCC      (b) Outdegree v.s. Clustering coefficient

Figure 9: The stucture of the DCC in follow network. Figure 9b plots the out-degree against $\bar{c}c$; Figure 9a shows $D_e$ of the DCC in follow network, the network is divided by the fractions of the total number of nodes.

**Densely Connected Core** In real-world online social networks, there exists a densely connected core (DCC) comprising between 1% and 10% of the highest degree nodes, such that removing this core completely disconnects the graph (Mislove et al., 2007). These high-degree nodes serve as hubs of the network, causing the network to become increasingly compact through the hub structure. As shown in Figure 9b, the nodes with higher degrees have significantly higher $\bar{c}c$ compared to other nodes. For these densely connected components, $D_e$ grows at a slow rate. In Figure 9a, we observe that the $D_e$ among the DCC of follow network is grows sublognitively. The network is generated by 5 rounds of simulation with 1e5 agents.

### B.3 MICRO-LEVEL EVALUATION

**Evaluation Metrics** In accordance with established evaluation metrics for graph generation Bergmeister et al. (2024), we report the maximum mean discrepancy (MMD) between the generated graphs and the test graphs, specifically focusing on degree distribution and clustering coefficient. Furthermore, we place particular emphasis on evaluating whether the degree distribution of the generated graphs conforms to a power law after expanding the graph to out-of-distribution sizes Clauset et al. (2009). To this end, we employ six key metrics in all:

**(1) MMD.D:** maximum mean discrepancy (MMD) of degree distribution between the generated graphs and the test graphs.

**(2) MMD.C:** maximum mean discrepancy (MMD) of clustering coefficient between the generated graphs and the test graphs.

**(3) MMD.S:** maximum mean discrepancy (MMD) of spectrum between the generated graphs and the test graphs Liao et al. (2019).

**(4) MMD.O:** maximum mean discrepancy (MMD) of node orbit counts between the generated graphs and the test graphs You et al. (2018).

**(5) $\alpha$:** The power-law exponent of the graph degree distribution.

**(6) $D_k$:** The Kolmogorov-Smirnov distance between the degree distributions of the generated and test graphs.

**(7) Valid:** Research demonstrates that degree distributions in complex networks are typically characterized by a power-law exponent $\alpha \in [2, 3]$ (Clauset et al., 2009). Accordingly, we define the validity measure for a graph as the proportion of graphs meeting the criteria $D_k < 0.1$ and $\alpha \in [2, 3]$. Set $k_{\min} = 2$ for the uniform calculation of the power-law fitness of both undirected and directed graphs.

**(8) GEM:** To quantify the level of structural alignment for the expanded graph, we establish the Graph Expansion Metric (GEM). Firstly, for the negative indicator MMD metrics, we utilize the transformation $1 - \frac{1}{1+e^{metric}}$, which maps the metrics to a range between 0 and 1. We then calculate the average of MMD and Valid metrics as GEM.

**Experiment Settings** Specifically, we sample a network dataset based on publication timelines to create our evaluation dataset. Since we only crawl for timestamp information of the CiteSeer and Cora datasets, these two datasets are used for our experimental evaluation. Following the timeline of graph evolution, we partition the network dataset into training and testing sets.

At a designated time point $t$, we filter the citation network using node timestamp information to obtain $G_{<t}$, which includes all nodes and edges prior to $t$. We sample small subgraphs from $G_{<t}$ to create a training set for deep learning methods and to generate the seed graph for GAG. The training set consists of sampled subgraphs with sizes ranging from 64 to 512 nodes, resulting in a train set comprising 160 subgraphs and validation sets comprising 32 subgraphs. For the test set, we filter the citation network for nodes and edges after $t$, denoted as $G_{>t}$. From $G_{>t}$, we sample large subgraphs of 1,000 nodes, resulting in a test set comprising 20 subgraphs.

[Revision: The existing graph generation methods have demonstrated promising results in generating small graphs, including works such as Vignac et al. (2023), Martinkus et al. (2022), and You et al. (2018). However, they have not explored the generation of out-of-distribution graph sizes, and the generated graph sizes are limited. To compare with traditional graph generation models, we must select methods that can expand beyond the training set graph sizes and efficiently generate large graphs. For rule-based graph generation methods, we set hyperparameters to ensure that the average degree of the expanded graph matches that of the seed graph. We adhere to the hyperparameters specified in the original papers for deep learning-based graph generation methods. All hyperparameter details are provided in Table 14.

]

**Ablation on Seed Graph Size** GAG distinguishes itself from traditional graph generation methods by generating graph data without requiring prior training. It achieves this through the simulation of human behavior, leading to the emergence of various structural features characteristic of real-world networks. Consequently, conventional graph evaluation methods cannot be applied. To address this, we design comparative experiments against existing graph generation models.

For this purpose, we select the Citeseer network as the seed graph and the training graph dataset for existing methods. We expand it to generate 20 distinct graphs with an additional 1000 nodes. We subsequently compare the structural metrics of these expanded graphs to those of subgraphs sampled from the Citeseer network of equivalent size. Specifically, we focus on whether the expanded graph structure exhibits power law characteristics typical of real-world network structures.

Additionally, we aim to investigate whether the size of the seed graph affects the validity of the final generated graph structure. To this end, we utilize the GAG to perform graph expansion on seed graphs of varying sizes. We plot the number of nodes in the expanded graphs against the corresponding values of $\alpha$. As shown in Figure 10, it is evident that larger seed graphs result in expanded graphs exhibiting higher values of $\alpha$. Furthermore, as the size of the expanded graphs increases, the $\alpha$ values gradually stabilize. This indicates that the GAG is capable of effectively and reasonably expanding graphs across different seed graph sizes.

Table 14: Training hyperparameters of baseline models. All unspecified hyperparameters default to their standard values.

| Model | Hyperparameter | Experiment |
|---|---|---|
| Erdös-Rényi (Erdos et al., 1960) | Linking propoblity | $\bar{k}/(|\mathcal{V}| - 1)$ |
| Barabási-Albert (Barabási & Albert, 1999) | Number of linking edges | $\bar{k}/2$ |
| Small-World (Watts & Strogatz, 1998) | Number of linking nodes | $\bar{k} \times 2$ |
| BiGG (Dai et al., 2020) | Ordering | DFS |
| | Accumulated gradients | 1 |
| | Batch size | 32 |
| GRAN (Liao et al., 2019) | Hidden size | 512 |
| | Embedding size | 512 |
| | Number of layers | 7 |
| | Number of mixtures | 20 |
| | Batch size | 16 |
| BwR (Diamant et al., 2023) | Model | GraphRNN (You et al., 2018) |
| | bw | 8 |
| | Hidden size | 128 |
| | Ordering | BFS |
| | Batch size | 32 |
| L-PPGN (Bergmeister et al., 2024) | Hidden embedding size | 256 |
| | PPGN embedding size | 128 |
| | Input embedding size | 32 |
| | Number of layers | 10 |
| | Number of denoising steps | 1024 |
| | Batch size | 32 |
| | EMA coefficient | 0.99 |
| | Number of spectral features | 0 |
| GraphMaker (Li et al., 2024a) | Variant | Sync |
| | Hidden size for timestep | 32 |
| | Hidden size for node | 512 |
| | Hidden size for node label | 64 |
| | NumberofMPNNlayers | 2 |
| | Learning rate | 0.001 |
| | Optimizer | AMSGrad |

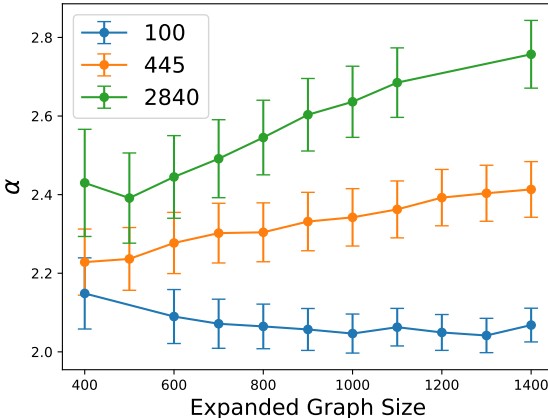

Figure 10: We present the results for seed graph sizes of 100, 445, and 2840, with the sizes of the expanded graphs plotted against their corresponding $\alpha$ values. Note that we only plot valid data $D_k < 0.1$.

**Supplementary Experiments** [Revision: To further demonstrate the reliability of the GAG framework, in addition to supplementing data from CiteSeer, we crawl for the necessary node attributes to enrich text attributes for Cora Sen et al. (2008). Following the experimental setup outlined in the paper, we designate Cora as the Seed Network and similarly compare it with existing graph generation models.

Table 15: Comparison with existing expansion-based graph generation models. For GRAN and GraphMaker, the generated graph degree distribution fails to converge when fitting a power-law distribution.

| | MMD.D↓ | MMD.C↓ | MMD.S↓ | MMD.O↓ | $D_k$ | $\alpha$ | Valid↑ | GEM |
|---|---|---|---|---|---|---|---|---|
| Cora | - | - | - | - | $0.07_{\pm 0.0}$ | $2.59_{\pm 0.01}$ | 1.00 | - |
| Erdös-Rényi | 0.25 | 1.41 | 0.54 | 0.27 | $0.13_{\pm 0.02}$ | $4.01_{\pm 0.17}$ | 0.00 | 0.29 |
| Barabási-Albert | 0.09 | 1.41 | 0.44 | 1.11 | $0.04_{\pm 0.01}$ | $2.4_{\pm 0.05}$ | 1.00 | 0.46 |
| Small-World | 0.60 | 1.41 | 0.50 | 0.20 | $0.14_{\pm 0.0}$ | $4.05_{\pm 0.02}$ | 0.00 | 0.28 |
| BiGG | 0.14 | 0.51 | 0.48 | 0.27 | $0.08_{\pm 0.01}$ | $3.19_{\pm 0.11}$ | 0.05 | 0.34 |
| GRAN | 0.15 | 0.50 | 0.55 | 0.28 | - | - | 0.35 | 0.40 |
| BwR | 0.32 | 0.23 | 0.34 | 0.19 | $0.1_{\pm 0.01}$ | $3.58_{\pm 0.08}$ | 0.00 | 0.35 |
| GraphMaker | 0.37 | 1.41 | 0.75 | 0.28 | - | - | 0.00 | 0.27 |
| L-PPGN | 0.15 | 0.92 | 0.32 | 0.59 | $0.06_{\pm 0.01}$ | $2.77_{\pm 0.04}$ | 1.00 | 0.50 |
| GAG | 0.35 | 0.84 | 0.41 | 1.21 | $0.09_{\pm 0.0}$ | $2.1_{\pm 0.01}$ | 1.00 | 0.47 |

As shown in Table 15, the expanded graphs generated by GAG adhere to a power-law distribution, with $\alpha = 2.1$. For MMD metrics, since GAG doesn't strictly enforce explicit graph structure constraints on the agents, its performance is not significantly better than other models, but it achieves comparable results. For the Valid metric, apart from GAG and the Barabási-Albert Model, the only existing deep-learning graph generation model capable of capturing the power-law distribution is L-PPGN. This demonstrates that L-PPGN is indeed capable of extrapolating to out-of-distribution graphs when trained on graphs that encompass the power-law distribution property. However, the unstable performance of L-PPGN across different datasets also highlights its sensitivity to the quality of the training dataset. In contrast, GAG, through human behavior simulation, can reliably generate graph structures that adhere to real-world network characteristics from the seed graph of varying sizes and quality. This illustrates not only the potential of LLM-based Agents in simulating human behavior but also underscores the reliability of the GAG framework. ]

## C EVALUATION OF TEXTUAL-ATTRIBUTE ALIGNMENT

### C.1 EXPERIMENT SETTINGS

We configure the complete Citeseer graph as $G$ and expand it to $G'$ with 5000 nodes. The processes involved are as follows:

**(1) F.shuffle:** This process randomly selects node-wise textual features from $G$ to create the node features of $G'$.

**(2) S.shuffle:** In this step, we randomly redefine the edges of $G'$, thereby disrupting the graph structure while maintaining the number of edges consistent with $\mathcal{V}'$.

**(3) SF.shuffle:** This refers to a combination of both F.shuffle and S.shuffle.

**(4) BWR+LLM:** Graph strcuture generated by Diamant et al. (2023). We give the LLM Citeseer as corpus and use it to generate textual features.

**(5) L-PPGN+LLM:** Graph strcuture generated by Bergmeister et al. (2024). We give the LLM Citeseer as corpus and use it to generate textual features.

**(6) BiGG+LLM:** Graph strcuture generated by Dai et al. (2020). We give the LLM Citeseer as corpus and use it to generate textual features.

**(7) BiGG+LLM:** Graph strcuture generated by Dai et al. (2020). We give the LLM Citeseer as corpus and use it to generate textual features.

**(8) GraphMaker+LLM:** Graph strcuture generated by Li et al. (2024a). We give the LLM Citeseer as corpus and use it to generate textual features.

**(9) GAG:** Graphs that are generated by the GAG framework.

## C.2 ABLATION STUDY OF LLM

To further illustrate the effectiveness of the generated graphs, we conduct an ablation study on the LLM for agent setup. We select four LLMs for this study: GPT-3.5-turbo, GPT-4o-mini OpenAI (2023) as top-ranking closed-source LLMs, and Llama-3-70BAI@Meta (2024) and Qwen2-72BYang et al. (2024) as top-ranking open-source LLMs.

Table 16: Ablation Study of LLM in GAG. Performance Comparison for Node Classification

| GNN | LLM | $\Delta ACC \downarrow$ | | | |
| | | SF.random | F.random | S.random | GAG |
|---|---|---|---|---|---|
| GAT | LLAMA. | $13.40_{\pm 3.52}$ | $24.11_{\pm 2.84}$ | $18.85_{\pm 2.55}$ | $\mathbf{2.26}_{\pm 1.19}$ |
| | GPT-3. | $12.30_{\pm 2.47}$ | $22.24_{\pm 2.86}$ | $16.69_{\pm 2.46}$ | $\mathbf{2.29}_{\pm 1.84}$ |
| | GPT-4o. | $12.03_{\pm 1.17}$ | $21.73_{\pm 2.91}$ | $15.57_{\pm 2.84}$ | $\mathbf{2.80}_{\pm 0.88}$ |
| | Qwen2. | $11.12_{\pm 2.53}$ | $15.84_{\pm 3.34}$ | $24.51_{\pm 2.28}$ | $\mathbf{10.50}_{\pm 1.94}$ |
| GCN | LLAMA. | $15.04_{\pm 1.99}$ | $23.20_{\pm 2.25}$ | $21.77_{\pm 2.04}$ | $\mathbf{3.61}_{\pm 1.32}$ |
| | GPT-3. | $13.55_{\pm 2.19}$ | $23.08_{\pm 1.39}$ | $20.58_{\pm 1.44}$ | $\mathbf{4.34}_{\pm 2.01}$ |
| | GPT-4o. | $14.56_{\pm 1.84}$ | $22.07_{\pm 2.25}$ | $20.82_{\pm 2.45}$ | $\mathbf{4.09}_{\pm 1.75}$ |
| | Qwen2. | $12.56_{\pm 1.65}$ | $15.37_{\pm 2.69}$ | $26.95_{\pm 2.16}$ | $\mathbf{10.05}_{\pm 1.87}$ |
| GCN2 | LLAMA. | $9.58_{\pm 0.88}$ | $9.33_{\pm 1.36}$ | $2.24_{\pm 1.58}$ | $\mathbf{0.49}_{\pm 1.50}$ |
| | GPT-3. | $9.07_{\pm 1.10}$ | $9.04_{\pm 1.80}$ | $1.43_{\pm 1.67}$ | $\mathbf{0.39}_{\pm 1.67}$ |
| | GPT-4o. | $8.89_{\pm 1.44}$ | $9.64_{\pm 1.10}$ | $1.19_{\pm 1.64}$ | $\mathbf{0.50}_{\pm 1.36}$ |
| | Qwen2. | $9.78_{\pm 1.48}$ | $\mathbf{9.19}_{\pm 1.44}$ | $11.40_{\pm 1.78}$ | $9.39_{\pm 0.93}$ |
| GraphSage | LLAMA. | $7.00_{\pm 0.92}$ | $7.13_{\pm 1.73}$ | $3.15_{\pm 1.84}$ | $\mathbf{0.09}_{\pm 1.74}$ |
| | GPT-3. | $6.72_{\pm 1.41}$ | $6.39_{\pm 0.92}$ | $2.35_{\pm 1.60}$ | $\mathbf{0.81}_{\pm 1.67}$ |
| | GPT-4o. | $6.97_{\pm 0.92}$ | $6.46_{\pm 1.61}$ | $2.05_{\pm 2.31}$ | $\mathbf{1.70}_{\pm 2.28}$ |
| | Qwen2. | $\mathbf{6.45}_{\pm 1.33}$ | $7.14_{\pm 1.68}$ | $12.73_{\pm 2.26}$ | $9.72_{\pm 2.43}$ |

As shown in Table 16, the LLM-based agents built on LLaMA2, GPT-3.5-turbo, and GPT-4o-mini are capable of generating networks that maintain the node and structural characteristics of the seed graph, thus ensuring effective performance transfer in downstream tasks. In contrast, Qwen-2 based agents do not guarantee performance transfer. We believe this is related to the ability of LLMs to emulate human behavior; Qwen-2 based agents fail to exhibit human-like creative behavior, resulting in less coherent generated graphs.

## D SCALABILITY OF GAG

[Revision: To further demonstrate the excellent scalability of the GAG framework, we conduct time measurements across various simulation scenarios. The tests are carried out on a computing machine equipped with 96 CPU cores and 376 GB of memory. For model inference, we utilized LLAMA-3-70B as the backbone LLM and employed the vLLM framework Kwon et al. (2023), running on a setup of four A-800 GPUs.

Table 17: The time cost (*hour*) of agents for one round of simulation.

| | N | P | T |
|---|---|---|---|
| CA | 5.01e+03 | 10 | 0.46h |
| RA | 3.91e+03 | 10 | 0.30h |
| SA | 9.97e+04 | 48 | 11h |

In accordance with the graphs generated for macro-level graph structure alignment experiments, we carry out CA simulation experiments for 200 rounds, RA simulation experiments for 33 rounds, and SA simulation experiments for 5 rounds. For each scenario, we measured the total simulation time and computed the average simulation time per round. These multi-round simulations provide a robust measure of the computational efficiency of the GAG framework. The results, summarized in Table 17, demonstrate GAG's capability to handle simulations with varying scales of agents. For simulation experiments with Thousands of agents, the average simulation time per round is 0.46 hours for CA experiments and 0.30 hours for RA experiments; For simulation experiments with Hundreds of thousands of agents, the average simulation time per round is 11 hours for SA. ]

## E    ABLATION STUDY OF S-RAG

To investigate whether the hyperparameter settings of S-RAG affect the generated network structure. We conduct ablation experiments on these hyperparameters. Given the variations in graph generation scenarios, we conduct the ablation experiments under the SA simulation. We run equal number of simulation rounds within the GAG to generate graphs.

In this section, we add a graph structure metric for measuring the proportion of the largest connect component within the network. We define the largest connect component of graph as LCC, so the proportion of LCC within the network is $|LCC|/|V|$. This aids us in comprehending the graph evolution progress.

### E.1    RECALL STAGE

In recall stage, the only hyperparameter is the number of searched items: $N_r$. Since the final number of documents interacting with the LLM is limited to $N_r$, we change $N_r$ and evaluate its impact on network structure.

Table 18: Ablation Study of $N_r$. The value of $N_r$ is proportional to $\bar{k}$ of the generated network.

| Network | $N_r$ | $|\mathcal{V}|$ | $|\mathcal{E}|$ | $\bar{c}c$ | $r$ | $|LCC|/|V|$ |
|---|---|---|---|---|---|---|
| Action | 3 | 9.47e+02 | 1.92e+03 | 0.07 | 0.10 | 0.03 |
| | 5 | 9.36e+02 | 2.20e+03 | 0.09 | -0.05 | 0.02 |
| | 10 | 9.58e+02 | 2.63e+03 | 0.09 | 0.02 | 0.05 |
| | 20 | 1.03e+03 | 3.03e+03 | 0.11 | -0.08 | 0.16 |
| Follow | 3 | 7.42e+02 | 1.27e+04 | 0.83 | -0.08 | 0.29 |
| | 5 | 7.39e+02 | 1.24e+04 | 0.81 | -0.06 | 0.44 |
| | 10 | 7.39e+02 | 1.29e+04 | 0.80 | -0.06 | 0.51 |
| | 20 | 8.91e+02 | 3.83e+04 | 0.82 | -0.18 | 1.00 |
| Friend | 3 | 7.42e+02 | 5.96e+03 | 0.88 | -0.13 | 0.25 |
| | 5 | 7.39e+02 | 5.76e+03 | 0.87 | -0.10 | 0.22 |
| | 10 | 7.39e+02 | 5.94e+03 | 0.89 | -0.10 | 0.24 |
| | 20 | 8.91e+02 | 1.83e+04 | 0.87 | -0.10 | 0.45 |

As shown in Table 18, we keep all other search parameters constant while varying the size of $N_r$. It can be observed that as $N_r$ increases, $\bar{k}$ of generated network also exhibits an upward trend. This trend is particularly pronounced in the follow network.

### E.2    RERANKING STAGE

To maximum the effectiveness of searched items, we implement the ReRanking stage in S-RAG. Initially, coarse ranking is performed to sort the searched items by their creator agent. Focusing on the *core* lable of creator. Subsequently, fine ranking is conducted based on the agent's individual preferences. We conduct an ablation study to explore the impact of different levels of personalization in ReRanking stage. And eventually its impact on the network structure.

We focus on the hyperparameters in the ReRanking stage, which mainly include: (1) Hub rate: $|HUB|/|V|$. (2) Attributes of $a_l$.

Table 19: Ablation Study of the hub rate ($|HUB|/|V|$). Higher hub rate contributes to the emergence of a large connected component.

| Network | $|HUB|/|V|$ | $|\mathcal{V}|$ | $|\mathcal{E}|$ | $\bar{cc}$ | $r$ | $|LCC|/|V|$ |
|---|---|---|---|---|---|---|
|        | 0.00 | 9.78e+02 | 2.64e+03 | 0.09 | -0.05 | 0.13 |
| Action | 0.10 | 1.02e+03 | 2.58e+03 | 0.09 | -0.07 | 0.10 |
|        | 0.20 | 1.03e+03 | 3.03e+03 | 0.11 | -0.08 | 0.16 |
|        | 0.00 | 7.79e+02 | 3.00e+04 | 0.84 | 0.02 | 0.63 |
| Follow | 0.10 | 7.82e+02 | 3.04e+04 | 0.84 | 0.05 | 0.63 |
|        | 0.20 | 8.91e+02 | 3.83e+04 | 0.82 | -0.18 | 1.00 |
|        | 0.00 | 7.79e+02 | 1.45e+04 | 0.89 | 0.21 | 0.27 |
| Friend | 0.10 | 7.82e+02 | 1.47e+04 | 0.88 | 0.29 | 0.43 |
|        | 0.20 | 8.91e+02 | 1.83e+04 | 0.87 | -0.10 | 0.45 |

**Coarse Ranking**  As shown in Table 19, an increase in the proportion of core users correlates with an upward trend in the proportion of the largest connected component within the network. Since the proportion of core users is increased, the likelihood of core users being searchable by general users is also increased, thereby fostering preferential attachment in the network. eventually, the proportion of the largest connected component within the network is increased.

**Fine Ranking**  To improving search algorithms based on personal preferences of agents, we design various filter items in fine ranking process, which are tailored to different simulation scenarios. The number of filter items is $N_f$. Within the SA simulation, filter items include: (1) Follow: Determines whether the content of the document is posted by an agent that the current agent follows. (2) Friend: Identifies whether the content of the document is was sent by an agent that is a friend of the current agent. (3) Topic: Assesses whether the content of the document is related to a topic that the current agent is interested in.

As illustrated in Table 20, $\bar{cc}$ of network increases as $N_f$ increases. Additionally, the impact level of different filter items is as follows: friend > topic > follow.

Table 20: Ablation Study of the fillter items used in fine ranking process.

| | Fillter Items | | | Network Structural Characteristics | | | | |
|---|---|---|---|---|---|---|---|---|
| Network | follow | topic | friend | $|\mathcal{V}|$ | $|\mathcal{E}|$ | $\bar{cc}$ | $r$ | $|LCC|/|V|$ |
|        | ✓ | - | - | 6.51e+02 | 1.88e+03 | 0.11 | 0.01 | 0.19 |
| Action | - | ✓ | - | 6.32e+02 | 1.82e+03 | 0.09 | -0.01 | 0.15 |
|        | - | - | ✓ | 1.02e+03 | 2.78e+03 | 0.09 | -0.05 | 0.12 |
|        | ✓ | ✓ | ✓ | 1.03e+03 | 3.03e+03 | 0.11 | -0.08 | 0.16 |
|        | ✓ | - | - | 6.51e+02 | 2.65e+04 | 0.78 | -0.19 | 0.92 |
| Follow | - | ✓ | - | 5.63e+02 | 2.03e+04 | 0.79 | 0.16 | 0.64 |
|        | - | - | ✓ | 7.70e+02 | 2.97e+04 | 0.84 | 0.19 | 0.69 |
|        | ✓ | ✓ | ✓ | 8.91e+02 | 3.83e+04 | 0.82 | -0.18 | 1.00 |
|        | ✓ | - | - | 6.51e+02 | 1.26e+04 | 0.83 | -0.13 | 0.46 |
| Friend | - | ✓ | - | 5.63e+02 | 9.74e+03 | 0.83 | 0.52 | 0.28 |
|        | - | - | ✓ | 7.70e+02 | 1.44e+04 | 0.89 | 0.21 | 0.31 |
|        | ✓ | ✓ | ✓ | 8.91e+02 | 1.83e+04 | 0.87 | -0.10 | 0.45 |

## F  CASE STUDY

[Revision:  Since GAG employs human behavior simulation for network generation, the process of connecting each network node to others closely mirrors real-world scenarios. This alignment enables a clear and interpretable understanding of the network evolution process.

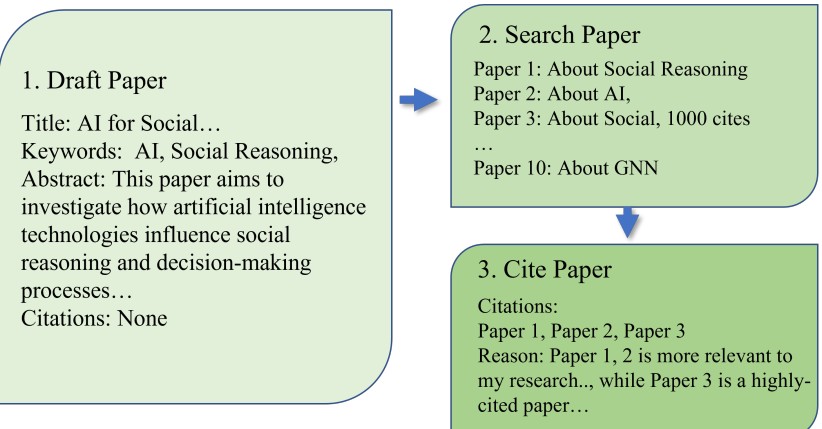

Figure 11: An illustration of the citation network evolution with LLM-based Agents.

To demonstrate the interpretability of our graph generation method, we present a case study using the CA simulation scenario. As illustrated in Figure 11, the formation of a citation network involves three primary steps. First, LLM-based agents collaboratively generate a paper draft through interaction and cooperation. Next, the agents search the corpus of stored papers within the environment to identify literature relevant to their research interests. For instance, query terms may correspond to research domains such as AI or social sciences.

Finally, after completing the search, the agents select and cite papers pertinent to their draft, providing explicit justifications for each citation. As illustrated in Figure 11, examples of such justifications include citing a paper due to its high citation count or its direct relevance to the research topic. Each citation edge in the network thus directly corresponds to an agent's citation action, offering a behavior-driven perspective on the graph construction process. This approach ensures that the graph generation process is inherently interpretable. ]

## G  HUMAN INTERFACE CONTROL

Previous work on employing LLMs for graph generation typically relied on predefined network structure features or a set of example networks Yao et al. (2024). Similarly, after understanding the reasons behind different structural characteristics of networks within GAG, we aim to enable users to control the entire simulation process by inputting prompts. This will guide and influence the various structural features of the final network.

To achieve this, we establish a control agent that accepts instruction frm users. Control agent transfers the instruction to a control profile for managing the simulation process of GAG.

As shown in Fig. 12,specifically, the hub rate controls the proportion of recommended core users, subsequently affecting the ratio of hub nodes in the network. The parameter $N_r$ determines the number of items recommended by the system, influencing the overall degree distribution. Additionally, parameter $N_f$ dictate the number of filter items in the ReRanking stage, impacting the network's clustering coefficient. Furthermore, the overall simulation time is adjusted by the number of agents $N$ per simulation round.

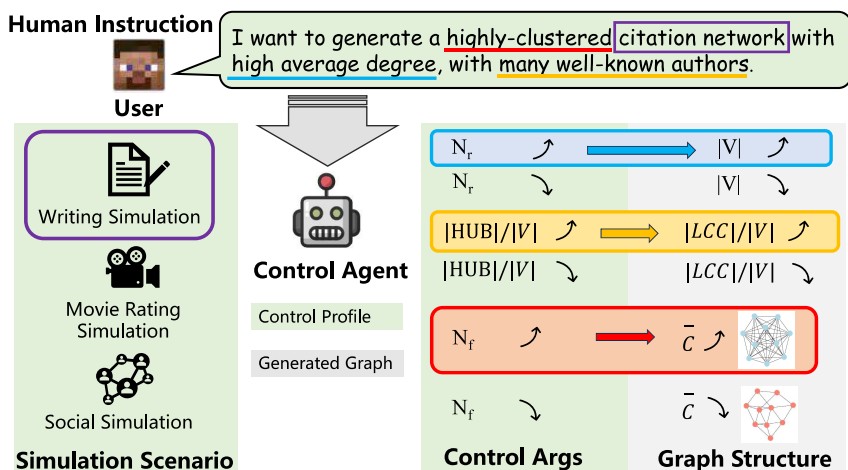

Figure 12: An illustration of the Control Agent in GAG Framework.

## H PROMPTS

Table 21: The prompt template of generating agent profiles for *CA*

I would like you to generate a series of random author's personal information.
These authors are interested in computer science, they are experts in various fields of CS.
I need you to give a list of author infos with the constraints for each attribute as follows:
(1) Name: Author's name
(2) Expertises: a list, The author's areas of expertises can be selected from the following areas:{expertises list}
(3) Institution: The author's institution, you can choose whatever institution you want, just give me one institution name
(4) Country: The author's country, you can choose whatever institution you want,just give me one country name corresponding to the institution
(5) Topics: a list, The topics this author is interested in, can be selected from the following topics:{topics list}
Here's some common used countrys you can infer to:
{countrys}
Please generate me a list of {author num} different authors, which can be loaded by eval function in python:
[{{
"name":"",
"expertises":[],
"institution":"",
"country":"",
"topics":[]
}},
...,
{{
"name":"",
"expertises":[],
"institution":"",
"country":"",
"topics":[]
}}]
Now please generate:

Table 22: The prompt template of generating agent profiles for *IA*

Your task is to give me a list of watcher's profiles. Respond in this format:
[ {
"gender": (F/M)
"age":(the age of the watcher)
"job":(the job of the watcher)
} ]
Respond:
Now please generate:

Table 23: The prompt template of generating agent profiles for *SA*

Your task is to give me a list of {num added} person's profiles for twitter users . Respond in this format: [ {{ "user name": "(str;The name of this user)", "user description":"(str;short and concise, a general description of this user, ordinary users or super  large users and the topics this person interested in)" }} ]
Now please generate:

