# OpenReview forum: "Large-Scale Dynamic Graph Generation via LLM-based Agent Simulation"
_ICLR.cc/2025/Conference — ICLR 2025 Conference Withdrawn Submission_

### Official Review · Reviewer_LQ7Q · 2024-10-31

**Soundness:** 2
**Presentation:** 2
**Contribution:** 2
**Rating:** 5
**Confidence:** 4

**Summary:**

This paper introduces GraphAgent-Generator (GAG), a generative framework for text-attributed graphs based on the interaction of multiple LLM agents using prompting and retrieval-augmented generation (RAG). Without any training, empirical studies demonstrate the effectiveness of GAG in scenarios like modeling the common structural patterns of real-world networks in network science. This approach also demonstrates competitive scalability, allowing generating networks of up to about 100K nodes.

**Strengths:**

**S1.** Using multiple LLM agents for graph generation is conceptually novel to the best of my knowledge.

**S2.** Generating graphs of up to 100K nodes demonstrates strong scalability of the approach.

**S3.** The authors consider multiple well-established structural patterns in network science for empirical studies.

**Weaknesses:**

**W1.** The paper primarily considers network science as the downstream scenario. While it may be of interest to the network science community, it's unclear whether there is sufficient interest from the ICLR community.

**W2.** In terms of problem formulation, the paper claims to consider the generation of dynamic and text-attributed graphs. However, the empirical studies primarily focus on static graph structural patterns, which is the same as various previous works, despite using different metrics. To justify the use of terms like "dynamic", I expect to see a setting for temporal network generation which requires modeling distribution shifts over time.

**W3.** The use of multiple agents is not sufficiently justified and discussed. With expressions like 100K nodes with 100K LLM-based agents at L82, it seems that each agent corresponds to exactly a node and likely agents only differ in prompts/prompt templates. This makes the whole idea a bit uninteresting and also leads to potential concerns for scalability. It will be more interesting if the agents exhibit drastically different functionalities. For example, in a heterogeneous graph, different agents can be responsible for different node types or edge types.

**W4.** Empirical studies fail to show that GAG can outperform models trained on specific datasets, making the selling point barely training-free graph generation. In that case, the authors need to spend more efforts in justifying why training-free models incapable of capturing fine-grained dataset-specific patterns is useful and desirable.

**W5.** The discussion and comparison with respect to previous related works is insufficient. To name a few:

- Watts & Strogatz. Collective dynamics of 'small-world' networks. Barabási–Albert has been proven to be non-ideal for capturing clustering coefficients while the Watts–Strogatz model, also a random graph model, performs much better in that case. The empirical studies about clustering coefficient should compare against the Watts–Strogatz model.

- Yoon et al. Graph Generative Model for Benchmarking Graph Neural Networks. This paper proposes the use of GNNs for graph generation evaluation.

- Li et al. GraphMaker: Can Diffusion Models Generate Large Attributed Graphs? This paper generates text-attributed graphs with up to 10K nodes and also employs GNNs for evaluation.

*Minor*
- The title in the PDF is different from the one in the OpenReview page.

- The citation at the end of L58 is not properly formatted.

- Notations should be more carefully handled for better readability, e.g., the Cs' in section 3.3.

- Table 7 in the appendix is not properly formatted.

**Questions:**

See Weakness.

---

> ### Author Response · Authors · 2024-11-20
> **Response to Reviewer LQ7Q**
>
> First, thank you very much for your comments and advices. We'll address your comments in the following.
>
> (1) Network science is a very broad field, and works on network science has been published in NIPS[1], WWW[2]. I think this kind of worry is unnecessary.
>
>
> (2) First, we have considered both visualization of dynamic network evolution, as has been detailed in Page 9: https://anonymous.4open.science/r/GraphAgent-2206/visualization/
> social network.mp4.
> Second, we also consider the evaluation of the dynamic evolution process of graph structure. As has been detailed in Section 4.1. Shrinking Diameter. Which is a surpursing property of graph evolution discovered in emperical studies. We have shown that the effective diameter of our generated graphs become smaller as the graph grows, successively replicating the shrinking diameter phenomenon.
>
>
>
> (3) The use of multiple agents aligns with our goal of generating **large-scale graphs**. Additionally, our proposed **N-ACTOR algorithm** relies on multiple agents to enable accelerated simulations. Using a single agent would not support large-scale simulations, making it infeasible to generate large-scale graphs.
>
> **GAG** is also capable of generating **heterogeneous graphs**. By defining different **mapping functions** (as detailed in Section 3.4), GAG can easily create heterogeneous graph structures. For example, in the **RA setting**, the Movie Rating Network (illustrated in Appendix A.3) includes a node set consisting of both **movie watchers** and **movies**.
>
> Furthermore, using agents to exhibit drastically different functionalities is indeed a valuable approach. We have incorporated this concept by employing a **control agent**, which creates a control profile to manage and oversee the simulation process in GAG.
>
>
> (4) For macro-level structural characteristic alignment, we examine three structural properties observed in real-world networks and demonstrate that they can be reliably replicated by **GAG**. This indicates that GAG, through human behavior simulation, effectively generates graph structures that align with real-world network characteristics. To further substantiate these findings, we have included additional macro-level structure alignment experiments in **Appendix B.2** as part of the empirical studies you mentioned.
>
> **But, we have already considered comparisons** with existing graph generation methods in terms of both **graph structure and textual features**:
>
> 1. **Graph Structure Comparison:**
>    Detailed experiments are presented in **Section 4.1**, and to further enhance the persuasiveness of our results, we have included supplementary experiments and ablation studies in **Appendix B.3**.
>
> 2. **Graph Textual Feature Comparison:**
>    Comprehensive comparisons are provided in **Section 4.3**, where we also expand the baseline graph generation models in the revised version. Additionally, to strengthen the evidence, we have included **Ablation Experiments on LLMs** in **Appendix C.2**.
>
>
>
> (5) For weakness 5.
> 1. **Adding More Random Graph Models as Baselines:**
>    We acknowledge the suggestion from other reviewers to include additional random graph generation models as baselines. In response, we have incorporated the small-world model as a baseline in **Sections 4.2** and **4.3**.
>
> 2. **GNN Benchmarking on Graph Generation:**
>    Thank you for the suggestion. We have added relevant references in **Section 4.3** to support GNN benchmarking in the context of graph generation.
>
> 3. **Inclusion of GraphMaker as a Baseline:**
>    We have included **GraphMaker** as a baseline for both the **Micro-level Structure Alignment** and **Textual-Attribute Alignment** experiments. However, it is important to note that while GraphMaker is capable of generating attributed graphs, it **cannot produce textual attributes** for graphs, which remains a unique strength of our proposed method.
>
>
> (6) For micro weaknesses. We change all the mentioned typos.
>
>
>
>
> [1] Sorscher B, Geirhos R, Shekhar S, et al. Beyond neural scaling laws: beating power law scaling via data pruning[J]. Advances in Neural Information Processing Systems, 2022, 35: 19523-19536.
>
> [2] Meusel R, Vigna S, Lehmberg O, et al. Graph structure in the web---revisited: a trick of the heavy tail[C] Proceedings of the 23rd international conference on World Wide Web. 2014: 427-432.

---

> ### Comment · Reviewer_LQ7Q · 2024-11-22
>
> Sorry for the late reply. Thank you for the detailed response and updated experiment results. To help me better assess the contributions and significance and also refresh my memory, I would greatly appreciate it if you could provide concise and precise explanations to the following questions, supported by specific references to existing literature or your experiment results whenever appropriate.
>
> 1. What are concrete downstream applications where text-attributed graph generation would provide practical value? I expect specific answers beyond an umbrella term like network science.
>
> 2. Your work proposes GAG as a training-free approach to graph generation. Could you:
>
>     - a) Identify the specific scenarios where a training-free approach offers advantages over training dedicated graph generative models
>
>     - b) Explain how your experimental setup reflects these target scenarios for both 1 and 2 a)
>
>     - c) Highlight the key results that demonstrate these benefits
>
> 3. Are training & inference efficiency/cost-effectiveness an important concern for the target scenarios? If so, what are the pros or cons of GAG?
>
> 4. Looking beyond the current work, what do you see as the most exciting possible directions for extending GAG that may inspire the graph ML/agent communities?

---

> > ### Author Response · Authors · 2024-11-23
> > **Response to Reviewer LQ7Q**
> >
> > Thanks for your kind reply. We'll address your questions as follows:
> >
> > (1) Text-attributed graphs (TAGs) have recently been a topic of interest in GNN [1], by combining graph structures with textual data, are frequently used in diverse real-world applications, including fact verification [2], recommendation systems [3], and social media analysis. Moreover, it serves as a foundational data source for build Graph Foundation model. However, for TAG datasets, including TAGLAS[4], InstructGraph[5], current methods rely on manual collection. Our proposed method is able to automatically generate TAGs, providing more flexibility and efficiency.
> >
> > (2)
> >    - a) A key limitation of current graph generation methods is their reliance on large-scale, high-quality training graphs. For example, to generate citation networks with \( 10^5 \) nodes, these methods require at least hundreds of citation networks of similar scale—a clearly infeasible task. **That's why these methods are often trained on graphs generated by BA, ER, and other random models for large-scale graph generation.**
> >
> >    - b)
> >       **Minimal Training Data Requirements**:
> >       - GAG requires only a small-scale seed graph with fewer than 1,000 nodes to generate graphs resembling real-world structures. Ablation studies (Appendix Figure 10) show that even with a seed graph of just 100 nodes, GAG can produce graphs with similar structural characteristics.
> >       - Remarkably, GAG can also generate graphs **from scratch**, without any initial graph dataset. An experiment demonstrating this capability for online social networks is included in Section 4.3. In contrast, existing deep-learning-based graph generative models cannot generate graphs from scratch.
> >
> >       **Generalizability to New Settings**:
> >       - Unlike existing methods, which fail to generalize to unseen settings, GAG can easily adapt by modifying the writing topics of LLM-based agents. For instance, GAG can generate citation networks for different domains (e.g., computer science, physics, medicine) by simply adjusting the agents’ topic focus.
> >
> >       **Superior Performance on Realistic Graph Generation**:
> >       - Using small-scale seed graphs, GAG generates graphs that more accurately capture real-world structural properties compared to existing methods. As demonstrated in experiment 4.1 (micro-level evaluation), GAG achieves a validation score of 1.0, outperforming deep-learning-based methods, which score 0 due to overfitting and failure to replicate the power-law distribution.
> >       - GAG surpasses the best baseline by 11% on GEM, demonstrating its effectiveness in simulating human behavior patterns and generating graphs that closely resemble real-world networks.
> >
> >       **Capability to Generate Text-Attributed Graphs**:
> >       - GAG can produce graphs with text attributes, enhancing their applicability. The effectiveness of these attributes is validated through GNN benchmarking tasks in experiment 4.2, a capability absent in existing graph generation methods.
> >
> >
> >
> >    - c) Overall benifits of GAG: 1. **Minimal Training Data Requirements**: GAG eliminates the need for large-scale graph datasets.  2. **Realistic Graph Generation**: The generated graphs resemble real-world networks, follow structural properties like power-law distribution, and include text attributes.
> >
> >
> >
> > [1] Yan H, Li C, Long R, et al. A comprehensive study on text-attributed graphs: Benchmarking and rethinking[J]. Advances in Neural Information Processing Systems, 2023, 36: 17238-17264.
> >
> > [2] Jie Zhou, Xu Han, Cheng Yang, Zhiyuan Liu, Lifeng Wang, Changcheng Li, and Maosong Sun. Gear: Graph-based evidence aggregating and reasoning for fact verification. arXiv preprint arXiv:1908.01843, 2019.
> >
> > [3] Jason Zhu, Yanling Cui, Yuming Liu, Hao Sun, Xue Li, Markus Pelger, Tianqi Yang, Liangjie Zhang, Ruofei Zhang, and Huasha Zhao. Textgnn: Improving text encoder via graph neural network in sponsored search. In Proceedings of the Web Conference 2021, pages 2848–2857, 2021.
> >
> > [4] Feng J, Liu H, Kong L, et al. TAGLAS: An atlas of text-attributed graph datasets in the era of large graph and language models[J]. arXiv preprint arXiv:2406.14683, 2024.
> >
> > [5] Wang J, Wu J, Hou Y, et al. InstructGraph: Boosting Large Language Models via Graph-centric Instruction Tuning and Preference Alignment[J]. arXiv preprint arXiv:2402.08785, 2024.
> >
> > [6] Zhu Y, Shi H, Wang X, et al. Graphclip: Enhancing transferability in graph foundation models for text-attributed graphs[J]. arXiv preprint arXiv:2410.10329, 2024.
> >
> > [7] Kong L, Feng J, Liu H, et al. Gofa: A generative one-for-all model for joint graph language modeling[J]. arXiv preprint arXiv:2407.09709, 2024.

---

> > > ### Author Response · Authors · 2024-11-23
> > > **Response to Reviewer LQ7Q**
> > >
> > > (3) Since simulation-based graph generation is a novel research direction, inference cost is not the primary concern. Instead, the focus is on exploring the capability boundaries of LLM-based agents in simulating human behavior patterns and generating graphs that closely resemble real-world networks.
> > >
> > >    - The main advantages: 1. The approach eliminates the need for large-scale training datasets. 2. GAG does not require any pretraining of graph generation models. 3. The framework generalizes effectively to large-scale graphs and unseen settings.
> > >
> > >    - The main disadvantages: 1. Generating *large-scale* graphs relies on LLM inference, which can be computationally expensive. But we have considered the time cost, and we employ the N-ACTOR parallelization strategy to accelerate the graph generation process, which effectively achieves a minimum speed-up of 90.4\% compared to without parallelization.
> > >    Additionally, as LLM technologies advance, simulation costs are expected to decrease further, enhancing the practicality of this approach.
> > >
> > > (4)
> > >    - As for the graph ML communities, the applications of GAG are broad and impactful. Since the collection of text-attributed graph (TAG) datasets heavily relies on manual collection, **all works requiring TAGs as training data can leverage GAG to perform graph data augmentation across domains.**
> > >
> > >       One of the most prominent areas is **Graph Foundation Models (GFMs)**. Models such as GOFA[7], InstructGraph[5], and GraphCLIP[6] heavily depend on high-quality TAGs for tasks like graph prompt tuning. By providing augmented TAG datasets, GAG ensures these models are trained on more diverse varietes of graph structures, enabling these models to learn more comprehensive and generalizable graph representations.
> > >
> > >    - As for the agent communites, our work explores the capability boundaries of LLM-based agents to simulate human behavior. While extensive prior works have employed LLM-based agents for role-playing, their evaluations have largely been limited to Turing tests[8] and human judgment[9]. In contrast, GAG evaluates simulation authenticity using the established theoretical framework of network science, offering a more rigorous and objective assessment of authenticity. Furthermore, the simulation scenarios in GAG are diverse and complex, providing a robust platform for testing agents in dynamic and realistic settings.
> > >
> > >       In the future, we plan to expand the range of simulation scenarios and systematically evaluate the graph structure quality produced by different LLM-based agents in GAG. We can effectively evaluate different LLMs from both role-playing and social science perspectives.
> > >
> > > [8] Shao Y, Li L, Dai J, et al. Character-LLM: A Trainable Agent for Role-Playing[C]//Proceedings of the 2023 Conference on Empirical Methods in Natural Language Processing. 2023: 13153-13187.
> > >
> > > [9] Ji J, Li Y, Liu H, et al. SRAP-Agent: Simulating and Optimizing Scarce Resource Allocation Policy with LLM-based Agent[C]//Findings of the Association for Computational Linguistics: EMNLP 2024. 2024: 267-293.

---

> > > > ### Comment · Reviewer_LQ7Q · 2024-11-25
> > > >
> > > > Thank you for preparing the responses as I requested. While these responses have not fully addressed my concerns, they help me think about how you may better pitch your work. In particular, LLMs are pre-trained over a huge amount of text data, with rich underlying structures and relations. Consequently, it may be promising to distill the knowledge of LLMs into structured representations like graphs. In particular, if your evaluations can prove that the generated graphs indeed lead to better performance of existing graph ML models, the empirical results will be very strong and convincing. I hope you can think more about the ideas I layout here. I've also adjusted my score to 5 in accordance.

---

> > > > > ### Author Response · Authors · 2024-11-26
> > > > > **Response to Reviewer LQ7Q**
> > > > >
> > > > > Thank you for your valuable feedback. However, proving that the generated tags can directly augment the performance of machine learning models is beyond the scope of this paper. It is well-established in the literature that text-rich graphs can significantly enhance ML model performance [1]. Our primary contribution lies in providing a valid, dynamic, text-rich graph generation framework. Therefore, we focus on ensuring that the generated graphs align with real-world text-rich graphs both in terms of structure and content.
> > > > >
> > > > > We hope this helps clarify the core motivation of our work, which is centered on graph generation, rather than the direct impact on ML model performance. We greatly appreciate your understanding and support as we continue to refine and present our approach.
> > > > >
> > > > > [1] Chen Z, Mao H, Li H, et al. Exploring the Potential of Large Language Models (LLMs) in Learning on Graph[C]//NeurIPS 2023 Workshop: New Frontiers in Graph Learning. 2023.

---

### Official Review · Reviewer_c4Qw · 2024-11-01

**Soundness:** 2
**Presentation:** 2
**Contribution:** 2
**Rating:** 3
**Confidence:** 4

**Summary:**

This paper addresses a synthetic graph generation problem by simulating human behaviors. The proposed framework, GraphAgent-Generator (GAG), uses LLM-based agents for graph generation. Through the iterative interactions between the agents, it extracts diverse graphs as collections of interaction data. In the experiments, the authors demonstrate that generated graphs follow multiple graph properties that real graphs exhibit.

**Strengths:**

- This paper proposes a graph generation method that simulates interactions among agents. Through the simulation, generated graphs are naturally dynamic and
- The authors generate large-scale graphs with 10 million edges and accelerate the graph generation process through parallel computing.
- The experiments demonstrate that the generated graphs are realistic in terms of both macro- and micro level evaluations.

**Weaknesses:**

- Certain parts of the paper lack clarity or contain ambiguous explanations.
    - In Section 1, the authors claim that interpretability in graph generation is one of the main contributions of this paper. However, I found no discussion about interpretability beyond the Introduction. Which experiments, if any, demonstrate that the graph generation is indeed interpretable, and why? In my view, the behavior of LLM-based agents is inherently difficult to interpret, as it heavily depends on the prompt. The authors should provide a detailed discussion on how interpretability is addressed, including case studies or qualitative analysis to support the claim.
    - In lines 378 and 379, the paper says “in Table 4, The generated networks exhibit small diameter De ∈[1.17,11.66], comparable to the range of [4,35] in real-world networks”. However, the De values of seven out of ten graphs in Table 4 are smaller than 4, which are not in the range of real-world networks that the authors cited. The authors should clarify why they claim these values are "comparable" given these results.
- Several experimental results are unclear. I wrote questions for the unclear parts in the Question section below (scalability/efficiency and ReRanking).

Comments:

- The main content of this paper is not self-contained. For example, Sections 3 and 4 lack the core technical information of the proposed approaches.
- $\mathcal{E}$ is defined twice in the paper: a set of edges and a set of embedding vectors.
- Some tables in Appendix are incomplete, e.g., Tables 7, 11, and 13 due to excessive width or inconsistent columns.

**Questions:**

- Which experiments or discussions demonstrate that the proposal is really interpretable (see Weakness about the interpretability of the method)?
- While the authors show the time cost (min) for generating one interaction data in Tables 5 and 6, I am also curious about the total time for generating the largest graph in this work. So, my question is: How long does it take to create the largest graph, e.g., Action and Friend in Table 4? Please include the experimental setting such as the numbers of agents and ports in your response.
- Why is the blue line (w.o. ReRanking) in Figure 3b zero across all the time?

---

> ### Author Response · Authors · 2024-11-20
> **Response to Reviewer c4Qw**
>
> First, thank you very much for your comments and advices. We'll address your comments in the following.
>
> (1) Regarding the interpretability of graph generation. Since we model the graph generation process as a simulation process, each added node and edge can be directly interpreted as paper writing, reference selection, retweet tweets and etc. Furthermore, we can directly gather reasons for paper referencing during the graph generation process. We provide a visualization of this process in Appendix. F.
>
> (2) Regarding Figure 3.b., upon removing ReRanking, $D_e$ stays at a ver small value across the graph evolution process, trending upwards from 2.6 to 2.96 (as mentioned in Line 395). We modify the legend in Figure 3.b.to clarify any misunderstanding.
> Moreover, to intuitively illustrate the graph evolution process modeled by GAG, we choose the SA simulation scenario and visualization in https://anonymous.4open.science/r/GraphAgent-2206/visualization/social_network.mp4
>
>
>
>
> (2) Regarding the small diameter of citation networks, in power-law models, this diameter is often expressed as $\log \log(n)$, where $n$ is the number of nodes. We incorporate empirical studies of diameter measurements to demonstrate that GAG can produce citation networks with a small diameter. Both the diameter range [4, 35] from empirical studies[1] and [1.17, 11.66] generated by GAG are relatively small when compared to the overall scale of their respective networks, making them comparable. To avoid misunderstanding, we clarify that while these ranges may differ numerically, they are both indicative of citation networks with similarly small diameters relative to their size.
>
>
> [1] Leskovec J, Kleinberg J, Faloutsos C. Graphs over time: densification laws, shrinking diameters and possible explanations[C] Proceedings of the eleventh ACM SIGKDD international conference on Knowledge discovery in data mining. 2005: 177-187.

---

> > ### Comment · Reviewer_c4Qw · 2024-11-23
> >
> > Thank you for clarifying some of my concerns raised in the review. Regarding the diameter, the diameter range of graphs generated by GAG might be comparable to the diameter range from empirical studies but is smaller. Do you have any idea about why GAG tends to generate graphs with smaller diameters? I am still not convinced and believe that the comparison of this kind of graph characteristics should be conducted after categorizing the graphs based on relevant factors/domains, allowing for a more detailed and meaningful analysis of the differences. Also, I am still curious about my second question that is related to the scalability.

---

> > > ### Author Response · Authors · 2024-11-23
> > > **Response to Reviewer c4Qw**
> > >
> > > Thanks for your kind reply. We'll address your questions as follows.
> > >
> > > a)
> > > The diameter of a graph has been an extensively studied topic. In random graph models, the diameter typically exhibits logarithmic or sub-logarithmic growth with graph size \( n \)[1, 5]. However, empirical studies reveal that, unlike random graph models, the effective diameter of many real-world networks actually decreases over time[1]. This indicates that logarithmic or sub-logarithmic growth provides only an upper bound for graph diameter growth.
> > > We present the effective diameters (\( D_e \)) of graphs generated by GAG in the following tables:
> > >
> > > |     |   Action |   Follow |   Friend |  Movie Rating |   User Projection |
> > > |:----|---------:|---------:|---------:|---------:|------------------:|
> > > | log($n$) |    11.51 |    11.51 |    11.51 |     8.33 |              8.27 |
> > > | log(log($n$))  |     2.44 |     2.44 |     2.44 |     2.12 |              2.11 |
> > > | $D_e$ |     2.79 |     2.85 |    11.66 |     2.98 |              1.17 |
> > >
> > >
> > > |     |   Citation |  Bib-Coupling |  Co-Citation |  Author Citation |  Co-Authorship |
> > > |:----|-------------------:|-------------------------:|--------------:|------------------:|----------------:|
> > > | log($n$) |9.34 |                     9.29 |          8.28 |              8.52 |            8.52 |
> > > | log(log($n$))  |               2.23 |                     2.23 |          2.11 |              2.14 |            2.14 |
> > > | $D_e$|               5.19 |                     2.94 |          3.89 |              3.44 |            5.77 |
> > >
> > >
> > > As shown in the tables, the diameters of GAG-generated graphs are bounded by log(\( n \)), further validating the presence of the **small-world property** in GAG-generated networks.
> > > The phenomenon of shrinking diameters has not been fully explained in the literature,and previous studies only provide degree densification power-law as a posiible explanation[1], the most relevant theoretical work[3] proves the existence of a constant upper bound for graph diameters but does not specify its value.
> > >
> > > Furthermore, **real-world graphs exhibit significant variability in diameters even within the same domain.** For instance:
> > > - Citation graphs: The diameter ranges from 5 to 10 for Arxiv citation networks but spans 10 to 35 for patent citation graphs[1].
> > > - Online social networks: Diameters vary widely, from 9 to 905 across different platforms.[4]
> > >
> > > b) About cost, we add more details about the cost in Appendix D.
> > >
> > > [1] Leskovec J, Kleinberg J, Faloutsos C. Graph evolution: Densification and shrinking diameters[J]. ACM transactions on Knowledge Discovery from Data (TKDD), 2007, 1(1): 2-es.
> > >
> > > [2] Coppersmith D, Gamarnik D, Sviridenko M. The diameter of a long‐range percolation graph[J]. Random Structures & Algorithms, 2002, 21(1): 1-13.
> > >
> > > [3] Lattanzi S, Sivakumar D. Affiliation networks[C]//Proceedings of the forty-first annual ACM symposium on Theory of computing. 2009: 427-434.
> > >
> > > [4] Mislove A, Marcon M, Gummadi K P, et al. Measurement and analysis of online social networks[C]//Proceedings of the 7th ACM SIGCOMM conference on Internet measurement. 2007: 29-42.
> > >
> > > [5] Coppersmith D, Gamarnik D, Sviridenko M. The diameter of a long‐range percolation graph[J]. Random Structures & Algorithms, 2002, 21(1): 1-13.

---

> > > > ### Comment · Reviewer_c4Qw · 2024-11-28
> > > >
> > > > Thank you for your kind reply. Regarding the scalability issue, if I understand correctly, do the SA simulation experiments require 11 hours per round for 5 rounds, totaling 55 hours? According to Table 17, the SA experiments include 100,000 nodes. However, considering the literature on text-attributed graphs [1, 2], where obgn-arxiv with 169,343 nodes is classified as a small graph, I believe that this graph generation is not truly “large-scale” as claimed in its title and content.
> > > >
> > > > [1] Hu, Weihua, et al. "Open graph benchmark: Datasets for machine learning on graphs." Advances in neural information processing systems 33 (2020): 22118-22133.
> > > > [2] He, Xiaoxin, et al. "Harnessing explanations: Llm-to-lm interpreter for enhanced text-attributed graph representation learning." arXiv preprint arXiv:2305.19523 (2023).

---

> > > > > ### Author Response · Authors · 2024-11-29
> > > > > **Response to Reviewer c4Qw**
> > > > >
> > > > > Thank you for your feedback. However, we would like to clarify that the benchmarks you referenced **rely on manual efforts for graph dataset construction**. If data collection is contingent on human labor, it becomes theoretically possible to generate graphs of any scale, provided sufficient resources are allocated to manual collection. This makes such approaches fundamentally different from automated graph generation methods.
> > > > >
> > > > > Our mention of **large-scale graphs is intended to highlight the differences in graph scale compared to existing graph generation models**. As detailed in Lines 475-482 of our paper, "current graph generation methods are primarily limited to generating graphs with up to 5,000 nodes, and specialized models designed for sparse large graphs can only produce simple grid-structured graphs with a maximum of 100,000 nodes." Furthermore, for the GraphMaker model[3], **even a graph with 13,000 nodes is considered large-scale**. In contrast, our GAG framework can generate  social graphs of **100,000 nodes with intricate small-world structures without assuming sparsity**. Importantly, our graphs are both **dynamic and text-attributed**, which is a significant departure from existing methods. While some models can generate dynamic and attributed graphs, none match the scale of GAG or support the generation of text-attributed dynamic graphs—both of which are key contributions of our work.
> > > > >
> > > > > To further clarify the distinctions between our approach and existing methods, we have included an updated comparison table below. We hope this will provide additional clarity and a better understanding of the unique capabilities of our method.
> > > > >
> > > > >
> > > > > |Name|Method|Text-Attributed|Attributed|Temporal|Scale(Nodes)|Year|
> > > > > |----|------|---------------|----------|--------|------------|----|
> > > > > |GAG|Simulation|√|√|√| 100000 |2024|
> > > > > |GraphMaker[3]|Difussion||√|| 13000 |2024|
> > > > > |L-PPGN[4]|Difussion||√|| 5037 |2024|
> > > > > |DiGress[5]|Difussion||√||88|2021|
> > > > > |Bwr-GraphRNN[6]|Auto-regressive|||| 5037 |2023|
> > > > > |BIGG[7]|Auto-regressive|||| 100000 |2020|
> > > > > |GRAN[8]|Auto-regressive|||| 5037 |2019|
> > > > > |GraphRNN[9]|Auto-regressive|||| 2025 |2018|
> > > > > |MDVAE[10]|VAE||√||<100|2022|
> > > > > |D-MolVAE[11]|VAE||||<100|2022|
> > > > > |GraphVAE[12]|VAE||√||<100|2018|
> > > > > |Mol-CycleGAN[13]|GAN||√||<100|2020|
> > > > >
> > > > >
> > > > >
> > > > > [1] Hu, Weihua, et al. "Open graph benchmark: Datasets for machine learning on graphs." Advances in neural information processing systems 33 (2020): 22118-22133. [2] He, Xiaoxin, et al. "Harnessing explanations: Llm-to-lm interpreter for enhanced text-attributed graph representation learning." arXiv preprint arXiv:2305.19523 (2023).
> > > > > [3] Li M, Kreačić E, Potluru V K, et al. Graphmaker: Can diffusion models generate large attributed graphs?, ICLR 2023.
> > > > > [4] Andreas Bergmeister, Karolis Martinkus, et al. Efficient and scalable graph generation through iterative local expansion. s.l., ICLR 2024; Vienna, Austria; May 7-11, 2024.
> > > > > [5] Clement Vignac, et al. DiGress: Discrete Denoising diffusion for graph generation, ICLR 2023.
> > > > > [6] Jang Y, Lee S, Ahn S. A Simple and Scalable Representation for Graph Generation[C], ICLR 2023.
> > > > > [7] Hanjun Dai, Azade Nazi, Yujia Li, Bo Dai, and Dale Schuurmans. Scalable deep generative modeling for sparse graphs. In International conference on machine learning, pp. 2302–2312. PMLR, 2020.
> > > > > [8] Renjie Liao, et al. Efficient graph generation with graph recurrent attention networks. Advances in neural information processing systems, 32, 2019.
> > > > > [9] Jiaxuan You, et al. Graphrnn: Generating
> > > > > realistic graphs with deep auto-regressive models. In International conference on machine learning, pp. 5708–5717. PMLR, 2018.
> > > > > [10] Yuanqi Du, Xiaojie Guo, Amarda Shehu, and Liang Zhao. Interpretable Molecular Graph Generation via Monotonic Constraints. In SDM, 2022. 6, 8, 20.
> > > > > [11] Yuanqi Du, Xiaojie Guo, Yinkai Wang, Amarda Shehu, and Liang Zhao. Small Molecule Generation via Disentangled Representation Learning. Bioinform., 2022. 18, 20
> > > > > [12] Martin Simonovsky and Nikos Komodakis. GraphVAE: Towards Generation of Small Graphs Using Variational Autoencoders. In ICANN, 2018. 1, 20
> > > > > [13] Łukasz Maziarka, Agnieszka Pocha, Jan Kaczmarczyk, Krzysztof Rataj, Tomasz Danel, and Michał Warchoł. Mol-CycleGAN: A Generative Model for Molecular Optimization. J. Cheminf., 2020. 7, 20

---

> > > > > > ### Comment · Reviewer_c4Qw · 2024-12-02
> > > > > >
> > > > > > Thank you for your response. While it addresses some of my concerns, it does not fully resolve them. The response highlights the proposal’s scalability compared to other highly unscalable graph generation methods. However, my primary concern lies in the practicality of the method for generating large-scale graphs, especially since the authors evaluated it on node classification tasks in the paper.

---

> > > > > > > ### Author Response · Authors · 2024-12-03
> > > > > > > **response to reviewer c4Qw**
> > > > > > >
> > > > > > > Thank you for your response. However, we would like to clarify the following points:
> > > > > > >
> > > > > > > 1. First, it is important to emphasize that the graphs in OGB[4] do not include dynamic text-attributed graphs. Although the ogbn-Arxiv dataset was subsequently augmented with text attributes in later works[5], it does not represent a dynamic graph. A more fair comparison should be made with DTGB[1], which consists of dynamic text-attributed graphs (DyTAGs). In DyTAGs, both nodes and edges are annotated with text descriptions, and both the graph structure and the text annotations evolve over time.
> > > > > > >
> > > > > > > Despite the broad applicability of DyTAGs, there is a notable scarcity of benchmark datasets specifically designed for DyTAGs, which hinders progress in many research domains. The only available benchmark is DTGB[1]. It is also worth highlighting that the median dataset size in DTGB is approximately 100,000 nodes, indicating that the graph generation scale of GAG is indeed practical for real-world applications.
> > > > > > >
> > > > > > > 2. Regarding concerns about the scalability of graphs generated by GAG, it is important to note that dynamic textual graphs are exceptionally rare and difficult to collect in real-world settings. As a result, we evaluate the applicability of the generated graphs by comparing them against existing dynamic graph benchmarks and text-attributed dynamic graph benchmarks.
> > > > > > >
> > > > > > > For dynamic graph benchmarks, one widely used dataset, EdgeBank[2], has a maximum of 13,169 nodes, which comfortably fits within the generation capabilities of GAG. Additionally, the recent TGBenchmark[3] dataset has a median node count in the hundreds of thousands, with approximately 5 out of 8 graphs containing around 10,000 nodes. GAG can effectively augment these existing benchmarks with larger-scale dynamic graph instances.
> > > > > > >
> > > > > > > Furthermore, for text-attributed dynamic graph benchmarks, DTGB[1] is currently the only available benchmark.  As previously mentioned, GAG's graph generation capabilities can significantly expand the scope of this benchmark, addressing the lack of scalability in current datasets.
> > > > > > >
> > > > > > > 3. Finally, as we have emphasized earlier, a fair evaluation of the scalability of a graph generation method should be conducted in comparison to existing graph generation approaches. Our results have demonstrated that GAG substantially outperforms current methods in terms of graph generation scale. Additionally, GAG generates graphs that are both text-attributed and dynamic, which is a key feature of our proposed method. Therefore, we believe that concerns regarding the scalability of our model should not be a significant issue.
> > > > > > >
> > > > > > >
> > > > > > >
> > > > > > > [1] Zhang J, Chen J, Yang M, et al. DTGB: A Comprehensive Benchmark for Dynamic Text-Attributed Graphs[J]. arXiv preprint arXiv:2406.12072, 2024.
> > > > > > >
> > > > > > > [2] Poursafaei F, Huang S, Pelrine K, et al. Towards better evaluation for dynamic link prediction[J]. Advances in Neural Information Processing Systems, 2022, 35: 32928-32941.
> > > > > > >
> > > > > > > [3] Huang S, Poursafaei F, Danovitch J, et al. Temporal graph benchmark for machine learning on temporal graphs[C] Proceedings of the 37th International Conference on Neural Information Processing Systems. 2023: 2056-2073.
> > > > > > >
> > > > > > > [4] Hu, Weihua, et al. "Open graph benchmark: Datasets for machine learning on graphs." Advances in neural information processing systems 33 (2020): 22118-22133.
> > > > > > >
> > > > > > > [5] He, Xiaoxin, et al. "Harnessing explanations: Llm-to-lm interpreter for enhanced text-attributed graph representation learning." arXiv preprint arXiv:2305.19523 (2023).

---

### Official Review · Reviewer_WTMu · 2024-11-03

**Soundness:** 3
**Presentation:** 3
**Contribution:** 3
**Rating:** 5
**Confidence:** 4

**Summary:**

This paper addresses the limitations of existing graph generation approaches, where rule-based models struggle to accommodate the complexity of real-world graphs, and deep learning methods face challenges in handling out-of-distribution scenarios. We propose an LLM-powered, agent-based simulation approach for graph generation to overcome these challenges. Our approach simulates interactions between agents based on their profile information, contributing to efficient and realistic graph generation through two key innovations:

**Relevance Filtering with RAG:** We leverage a Retrieval-Augmented Generation (RAG) framework to filter out irrelevant agents, ensuring that only pertinent agents are included in the simulation.

**Parallel Processing with Community Structure:** By using parallel processing techniques and incorporating community structure, we accelerate the generation process, making it feasible to handle large-scale, complex graphs.

**Strengths:**

(1) The proposed LLM-based agent for graph generation is a promising direction for social network simulation, especially for large-scale networks.

(2) The proposed textual-graph generation and parallel generation to simulate a very large-scale network in a very short time are very important and novel problems to address in this type of graph generation.

(3) Extensive experiments are conducted to verify the effectiveness and efficiency of this work.

**Weaknesses:**

(1) Some of the claims in this paper are not so accurate in terms of introduction and related work. See questions (1), (2), (3)

(2) Using RAG to filter irrelevant candidates when generating social interactions based on textual similarity would limit the scope of the generated network to a homophily network. For example, taking dating networks as an example, if we only use gender information to retrieve, we might generate a lot of people sharing the same gender interacting with each other, which might not align well with real-world cases.

(3) The experiment setting for node classification is very weird. Firstly, the generated graph expands upon the original graph; why can't we directly generate the new graph and compare its node classification performance with the original one? Secondly, the baseline is very weak; the performance would significantly drop if we randomly select a node feature to create a node feature or shuffle. It might be better to compare with some better baselines, such as the ones that can maintain network homophily.

**Questions:**

(1) For deep-learning-based graph generative models, although it can only fit training graphs, why could this be a problem that motivated the investigation of LLM-based agent simulated graph generation?

(2) The motivation of this work, in terms of how existing graph generation models have problems, is a little bit inaccurate. . For example, several existing graph diffusion models [1] have demonstrated the capability to generate community structure (which is a macro property), which does not closely align with the claim in line 53, "struggle to capture macro properties." Furthermore, the claim in line 50, "when generating larger graphs beyond the size of the observed dataset," is inaccurate. It treats the graph generation as an OOD problem. However, the proposed method in this paper has also not been verified by the OOD experiment.

[1] Limnios, Stratis, et al. "Sagess: Sampling graph denoising diffusion model for scalable graph generation." arXiv preprint arXiv:2306.16827 (2023).

(3) In line 100, the maximum size of the expanded graph is limited to 144 nodes. Actually, the recent scalable graph diffusion models could handle social networks up to thousands of nodes [2]. However, I think this is far less than the real-world social network, which can still motivate this proposed work. However, if only saying 144 nodes, it is still slightly inaccurate.

(4) In related work, a very important work "Chang, Serina, et al. "LLMs generate structurally realistic social networks but overestimate political homophily." arXiv preprint arXiv:2408.16629 (2024)." is missing. It is better to include some discussions over here.

(5) In line 770, how do you crawl the node attributes to enrich text information for Citeseer? I checked the reference thereafter, but it is merely a paper collecting the Citeseer dataset. It might be better to provide more details on obtaining real-world textual profile information to initialize the agent.

(6) In line 199, why the C only considers one neighbor of each node $E_i = \{(v_i, v_j)|v_i, v_j \in \mathcal{V}\}$.

(7) RAG seems mainly based on textual similarity to pre-filter, which might only apply to homophily social networks. Is there any idea how it can be generalized to a heterophily network, e.g., a dating network? In this case, it might be difficult to retrieve based on gender information.

(8) How much cost does it take for one round of simulation?

---

> ### Author Response · Authors · 2024-11-20
> **Response to Reviewer WTMu**
>
> First, thank you very much for your acknowledgement and advices. We are very much appreciate it. We'll address your comments in the following.
>
> (1) For question 1 and 2.
> Indeed, we acknowledge that existing graph generation models can capture macro properties, but only if **they're given high-quality training graphs.**
>
> - First, for existing graph generation models, we need to **collecting the necessary large-scale graphs** as datasets. Existing models typically require at least 50 graphs to train model. To generate citation networks with 10^5 nodes, this would require collecting at least 50 citation networks of similar scale—a clearly infeasible task. **In contrast, GAG can simulate the evolution process of citation networks from small to large scales.**
> As demonstrated in Experiment 4.2, the citation networks generated by GAG adhere to both the micro-level and macro-level properties observed in real-world citation networks. Existing deep learning-based graph generative models, however, struggle to transfer graph structures learned from small-scale citation networks to larger ones. This limitation is evident in their failure to reproduce the power-law distribution in generated graphs. **Even in the absence of any graph datasets, GAG is able to generate citation graphs from scratch.**
> We incoporate one example of generating online social networks from scratch, as shown in Section 4.3. However, existing deep-learning-based graph generative models cannot generate graphs from scratch.
>
> - Additionally, these models cannot **generalize to new settings where networks have not yet been observed.** For instance, to generate citation networks in different domains, GAG requires only a change in the writing topics of LLM-based agents, **easily generating citation networks for topics of computer science, physics, medicine, etc.**
>
> In summary, to generate large-scale citation networks with existing graph models, we would need to gather many large-scale citation networks at first. And the models cannot be generalized to new settings where networks are not yet observed.
>
> (2) For question 3, we have added more discussions about recent scalable graph diffusion models.
>
> (3) For question 4, we have added more discussions about "LLMs generate structurally realistic social networks but overestimate political homophily." in
> Section 2. This paper provide valuable results for evaluating political homophily of generated graphs, but it's limited to the social networks. Our framework is applicable to other domains, such as citation networks, movie networks, etc.
>
> (4) For question 5, we crawl for the author information, abstract, citations and etc. for the CiteSeer and Cora Dataset. The Dataset is open-sourced in the GitHub repository.
>
> (5) For question 6, we transform the node representations into textual format. We consider all neighbouring edges of node $v_i$, to enrich the tetual information of node $v_i$. This is detailed in Appendix.A.2.
>
> (6) S-RAG retrives the most relvent environment information to the queries sent by agents, it doesn't necessarily mean GAG only retrieves homophily social networks. The foundational idea behind GAG is that social networks consist of two types of entities: **actors** and **items**, that are related by affiliation of the former in the latter. This interaction can naturally be modeled as a **bipartite graph**, denoted as $ B(Q, U) $, where $ Q $ represents the actors and $ U $ represents the items. In GAG, **LLM-based agents** form the set $ Q $, consiting of $ N $ agents to simulate diverse actor roles, such as authors, movie watchers, or tweet users. Meanwhile, the $ U $ set, representing items, is initialized with $ n $ entities, such as papers, tweets, or movies, depending on the simulation setting. Through continouse simulation rounds,
> the bipartite graph $ B(Q, U) $ evolves with generated interaction data.
>
> For each simulation round, each author in $Q$ is provided with partial environmental information through S-RAG. The partial environmental information denotes the textual representation of subgraph from $U$.
> Authors are prompted with action template $T_a$ to form interaction data,  which include new item nodes, which are added to $U$; as well as actor-item links. For example, in citation networks, authors can write a new paper and cite other papers. The newly generated links and items are added to $C$, where changes in $C$ correspond to the evolution of $ B(Q, U) $.
>
> The flexibility of GAG lies in its ability to model not only homophilous networks but also heterophilous ones. To achieve this, we modify the mapping functions to fold $ B(Q, U) $ into graph $G$ that capture heterophily.
> For example, paper-author bipartite graph can be folded into citation graphs, co-authorchip graphs and etc.
> The details of these mapping functions are described in Section 3.4, with further implementation specifics provided in Appendix A.3.
>
> (7) About cost, we add more details about the cost in Appendix D.

---

> > ### Comment · Reviewer_WTMu · 2024-11-26
> > **Thank you for your response**
> >
> > Thank you for your detailed response.
> >
> > **(1) Motivation:**
> > I am still a little bit unconvincing about the motivation here. I am unconvinced by the OOD issue of deep-learning-based graph generative models in simulating social networks. If there is a distribution shift problem, the same thing happens with the agent-based method. Just like the author mentions, you can change the writing topic. This is based on the assumption that the network connections depend on the network features.
> >
> > However, I do think the point that requiring more training graphs and also the scalability issue is indeed a good motivation to inspire research in agent simulation for large-scale social networks.
> >
> > **(2) Scalability Issue:**
> > Question 3 is not sufficiently addressed. The author uses scalability issues to motivate the agent simulation. However, this paper tried only 144 nodes, which does not support your motivation.
> >
> > **(3) Cost Issue:**
> > I checked the updated Appendix D, but I did not find the cost for calling GPT-4o and only found the time. Could the author also provide that information?

---

> > > ### Author Response · Authors · 2024-11-26
> > > **Response to Reviewer WTMu**
> > >
> > > Thanks for your kind reply. We'll address your questions as follows:
> > >
> > > (1) About Motivation. The reason GAG outperforms existing graph generation models in addressing out-of-distribution (OOD) issues lies in the assumption that LLM-based agent behavior closely mirrors human behavior patterns. Specifically, **human preferential attachment behavior** induces **power-law distributions in network structures**. Since LLM-based agents can **replicate this behavior**, the expanded network naturally adheres to a power-law distribution. This assumption is supported by concurrent work [2,3].
> > > The ability of **LLM-based agents to effectively simulate human behavior** is the central motivation behind our work. By mimicking the underlying graph generation mechanism, LLM-based agents ensure that the expanded network retains key structural characteristics of real-world networks, such as the power-law distribution.
> > >
> > >
> > > In contrast, deep learning-based graph generation models cannot learn the underlying graph generation mechanisms from the observed graph datasets. If these datasets do not exhibit a power-law distribution, the generated graphs will simply replicate the connectivity patterns of the observed data. As a result, the generated networks will fail to capture the true structural characteristics of real-world networks.
> > >
> > > (2) Scalability Issue: I'm afraid there is a significant misunderstanding regarding the scale of the generated graphs. The scale of nodes and edges for the different generated graphs is as follows:
> > >
> > > | | Citation| Bib-Coupling| Co-Citation| Author Citation| Co-Authorship|
> > > |:----|---------:|---------:|---------:|---------:|------------------:|
> > > |  $ \|\mathcal{V}\| $   | 1.14e+04| 1.09e+04| 3.93e+03| 5.01e+03| 5.01e+03|
> > > |  $ \|\mathcal{E}\| $  | 3.63e+04| 1.22e+07| 3.27e+04| 2.41e+05| 2.08e+04|
> > >
> > > | |Action| Follow| Friend| Movie Rating| User Projection|
> > > |:----|---------:|---------:|---------:|---------:|------------------:|
> > > |  $ \|\mathcal{V}\| $   | 9.97e+04| 9.96e+04| 9.96e+04| 4.17e+03| 3.91e+03|
> > > | $ \|\mathcal{E}\| $         | 9.07e+05| 1.53e+06| 5.01e+05| 3.25e+04| 9.04e+05|
> > >
> > > A detailed analysis of the graph generation scale is provided in Section 4.3. We refer to Bigg [1] for their work on graph generation beyond the training distribution through extrapolation, where they use planar and tree datasets to generate graphs with 48 to 144 nodes. However, their approach is limited to graphs with a maximum of 144 nodes. In contrast, GAG is capable of expanding small-scale graphs (with hundreds of nodes) to thousands of nodes. Notably, the largest graph generated by GAG in the action network contains nearly 100,000 nodes. I’m afraid there has been a misunderstanding regarding the source of the data.
> > >
> > > (3) About cost.
> > > As detailed in Appendix A.1 (line 831), we use an open-source model for macro-level structure alignment (i.e., graphs in Table 4). In response to your request for the cost of generating the largest graphs, we have incorporated the cost details of LLAMA in Appendix D.
> > > For closed-source LLMs, we can estimate the theoretical cost as follows:
> > > $$
> > > \text{Inference call number} = \frac{\text{Round simulation time}}{\text{interaction data generation time (LLAMA-based simulation, \( P = 24 \))}}
> > > $$
> > >
> > > Based on an input of 2000 tokens and an output of 500 tokens, the detailed costs for one simulation round in different scenarios are provided in the table below:
> > >
> > > | | N | Cost(GPT-3.5-turbo)|Cost(GPT-4o-mini)|
> > > |:------------------|---------:|---------:|---------:|
> > > | CA | 5.01e+03 | 27.27\$| 1.818\$|
> > > | RA | 3.91e+03 | 299.7\$  | 19.9\$|
> > > |SA | 9.97e+04 |  990\$ | 66\$|
> > >
> > > For large-scale graph generation, the simulation cost is substantial, which is why we opt for open-source LLM models.  Please note that our parallel acceleration does not **reduce the number of LLM inference calls**; rather, it improves the efficiency of inference calls and reduces the idle waiting time for llm-based agents.
> > >
> > >
> > >
> > > [1] Bergmeister A, Martinkus K, Perraudin N, et al. Efficient and scalable graph generation through iterative local expansion[J]. arXiv preprint arXiv:2312.11529, 2023.
> > >
> > > [2] Li X, Xu Y, Zhang Y, et al. Large Language Model-driven Multi-Agent Simulation for News Diffusion Under Different Network Structures[J]. arXiv preprint arXiv:2410.13909, 2024.
> > >
> > > [3] Papachristou M, Yuan Y. Network Formation and Dynamics Among Multi-LLMs[J]. arXiv preprint arXiv:2402.10659, 2024

---

### Official Review · Reviewer_8DUN · 2024-11-03

**Soundness:** 2
**Presentation:** 2
**Contribution:** 3
**Rating:** 5
**Confidence:** 2

**Summary:**

This paper presents a method for generating dynamic graphs via agent-based simulation. Graphs considered in this work contain text-based attributes and agents are modelled through Large Language Models, a method that has been shown to produce simulations of human-like behaviour. Rather than focusing solely on the final static graph, this method allows simulating the dynamic, organic growth of a complex network. In addition, using an LLM as the center-piece for the graph generating process allows the authors to generate both the graph topology and textual attributes of the graph.

The authors test the generated graphs with network science based metrics, demonstrating power law behaviour, small world phenomenom and preferential attachment (through shrinking diameter) in the generated networks. The authors also show that generated graphs retain properties useful for downstream tasks, such as node classification. Finally, the generated graphs are shown to scale up to 10 million edges and almost 100,000 nodes.

**Strengths:**

- **Method**: the method presented is an interesting take on graph generation, particularly the ability to generate both network structure and text attributes simultaneously, which naturally enhance the realism of the generated networks.
- **Large-scale**: even if the *large* label can be questioned ($10^5$ nodes is on the lower end of large scale), the paper presents a significant advancement to the size of networks generated by deep-learning + agent based methods.

**Weaknesses:**

- **Presentation**: ignoring the multitude of typos, which are easily fixed for a camera-ready version, I think the paper does itself a disservice in the way it presents its major strengths. It is clear to me that the content of the contribution was too much to fit in a 10 page paper and the final output omits too much detail to the point where it is hard to understand what was actually done. Sections 3.4, 3.5 and 4.2 are the best examples of this issue.
- **Experimental setup**:
   - **Experiment 4.1**: the random network models the authors choose to compare with are too simple. There is wide literature of random networks that approximate characteristics like small-world behaviour (Watts-Strogatz, for example, albeit not dynamic) or combination thereof, so comparing the properties of generated graphs against a random network model like Erdos-Renyi is naturally going to favour the proposed model.
   - **Experiment 4.2**: there is a lack of comparison of the quality of generated graphs against other graph generation models in the literature. Even if other models were unable to generate both topology and text features simultaneously, a more convincing experiment would be generating the graph topology using other methods, generating the text features with an LLM based on the CiteSeer corpus and demonstrating that GAG is better than this mishmash.

**Questions:**

- Have you considered using network comparison or network alignment to verify the quality of the networks you generated, even ignoring text features?
- Network motifs encode functional behaviour within the network and are extremely powerful at distinguishing networks with similar generation mechanisms, are motif fingerprints and graphlet distributions of real networks replicated by your graph generation method?

---

> ### Author Response · Authors · 2024-11-20
> **Response to Reviewer 8DUN**
>
> First, thank you very much for your acknowledgement and advices. We are very much appreciate it. We'll address your comments in the following.
>
> (1) Regarding presentations of the technical details of this paper. Given the 10-page limit for this paper, some trade-offs were necessary in presenting technical details.
>
> In **Section 3.4**, we provide additional details about the network evolution process in **Appendix A.2** and **Appendix A.3**. The foundational idea of our model is that social networks comprise two types of entities: **actors** and **items**, connected through the affiliation of the former with the latter. This interaction is naturally represented as a **bipartite graph**, denoted as $ B(Q, U) $, where $ Q $ represents the actors and $ U $ represents the items.
>
> In the context of **GAG**, **LLM-based agents** constitute the set $ Q $, consisting of $ N $ agents simulating diverse actor roles, such as authors, movie watchers, or tweet users. Meanwhile, the set $ U $, representing items, is initialized with $ n $ entities such as papers, tweets, or movies, depending on the simulation setting.
>
> To clarify the **pairwise interaction process**, we use the citation network as an example. In a citation network, the bipartite graph comprises **authors** ($ Q $) and **papers** ($ U $). The **environment corpus** ($ C $) records the interactions between $ Q $ and $ U $, encapsulating their dynamic relationships. To initialize the environment corpus $ C $, each node in $ U $ is transformed into a textual format using an **environment template** ($ T $). For instance (More examples available in Appendix.A.2.):
>
>     Node Features: Academic Paper
>     Title: <Title>
>     Topic: <Topic>
>     Abstract: <Abstract>
>     Author(s): <Author(s)>
>     Edge Features: Citation relationships for paper node.
>
> Our motivating example is a “social” network that emerges from search engine queries. For each simulation round, each author in $Q$ is provided with partial environmental information through queries. And author are prompted with action template $T_a$ to form new items, which are added to $U$; as well as links between $Q$ and $U$. For example, in citation networks, authors can write a new paper and cite other papers. The newly generated links and items are added to $C$.
> For instance (More examples available in Appendix.A.3.):
>
>     Action Template
>     Agent Profile: <Agent Profile\>
>     Agent Memory: <Agent Memory\>
>     Environment Observation: Partial Environmental Feedback provided by S-RAG, denoted as $o_l$. Abstract information for searched papers.
>     Human Instruction: <Instruction>...
>     Now write a version of your paper and cite the papers you need to cite.
>     ------------------------------------------------------------------
>     Respond Example
>     Node Features: Academic paper. Title, Keywords, Abstract.
>     Edge Features: Paper references.
>
>
> - For section 3.5. We need to obtain distinct types of graphs. Since after $k$ simulation rounds, the bipartite graph $B(Q,U)$ evolves, which gradually enriches $C$ into $C'$. Thus we define a range of mapping functions based on edge and node types
> to extract distinct graphs from $C'$. We incroporate the mapping functions in Appendix.A.3.
>
>
> - For section 4.2, We add additional details about the GNN experiments in the Appendix.C.1, and add more details about the evaluation metrics in Section 4.2.
>
>
> (2) Regarding adding more baselines for random networks. We add more baselines both in section 4.1. (Micro-level Evaluation). We add more datasets in Appendix.B. in the revision version.
>
> (3) For structure alignment, we have done extensive experiment on both macro-level and micro-level properties of graph structure and we ignores text features.
> For macro-level, we focus on the power-law, small-world, shrinking diameter and other stucture properties.
> For micro-level, we focus on the degree-distribution, clustering coefficient and other micro-level properties.
> As detailed in section 4.1. and more details in Appendix.B. For both macro-level and micro-level structure alignment.
>
> (4) About question 2, We generate the graph topology using existing graph generation methods, coupled with the text features generated with an LLM based on the CiteSeer corpus. We add the experiment details in Appendix.C. The experiment results are shown in Table 3 in Section 4.2. in the revision version.
>
> (5) Indeed, network motifs and graphlets are ideal indicators for graph generation. Following [1], we add additional metrics for micro-level evaluation of graph structure. We add additional experiments in Table 2 and Table 15.
>
> [1] Diamant N L, Tseng A M, Chuang K V, et al. Improving graph generation by restricting graph bandwidth[C] International Conference on Machine Learning. PMLR, 2023: 7939-7959.

---

### Official Review · Reviewer_kyGp · 2024-11-04

**Soundness:** 3
**Presentation:** 3
**Contribution:** 3
**Rating:** 6
**Confidence:** 3

**Summary:**

The paper introduces GAG, a framework for generating large-scale human interaction graphs. The framework includes an S-RAG algorithm to simulate human interactions and an N-ACTOR parallel processing component. The paper shows that GAG preserves real-world structural characteristic.

**Strengths:**

- The paper effectively shows that GAG generates graphs that mathc structural characteristics found in real world graphs.
- GAG's frameweork of generating graphs via simulating human interactions using LLM-based agents is novel.

**Weaknesses:**

- The paper only evaluates the micro-level structural characteristics of GAG on the Citeseer dataset. This greatly limits the scope of the evaluation, and the authors should expand this evaluation to other datasets across domains to increase the generalizability of the results.

**Questions:**

- In Section A.1, the authors mention that "for the LLM backbone, we have chosen open source model of Llama-3 AI@Meta (2024)". Which Llama 3 model was chosen?

---

> ### Author Response · Authors · 2024-11-17
> **Response to Reviewer kyGp**
>
> For macro-level structure alignment, we chose LLama-3-70B, while GPT-3.5-turbo was selected for micro-level alignment. In the ablation study of LLMs, we also included GPT-4o-mini and Qwen-2-72B. These details have been incorporated into the revised version for clarification.

---

> ### Author Response · Authors · 2024-11-20
> **Response to Reviewer kyGp**
>
> The experiments for micro-level structure alignment requires text-rich dynamic graphs, which are highly scarce. To address this limitation, we enhanced the original Cora dataset by adding timestamps and text-rich attributes. The enriched datasets are available in our open-source GitHub repository. Additionally, we conducted further experiments on additional datasets, as detailed in **Appendix B.3 (Table 15)**.

---

> ### Comment · Reviewer_kyGp · 2024-11-27
> **Response to Author Comment**
>
> Thank you for your response. All my comments have been addressed, and I've increased my score as a result.

---

### Author Response · Authors · 2024-11-25
**Response to Reviewers**

We sincerely appreciate the time and effort all the reviewers have dedicated to evaluating our work. As we approach the end of the rebuttal process, I would greatly appreciate any comments or suggestions. If possible, please share your response at your earliest convenience so that we can further discuss the paper content. Your support and insights are valuable to us. Thank you once again for your assistance.

---

### Author Response · Authors · 2024-11-27
**Response to Reviewers**

To clearly indicate the modifications made to the original paper, we have highlighted all changes in blue in the revised version, which will facilitate the reviewers' assessment of the experimental results. We sincerely appreciate all the relevant comments provided by the reviewers. However, as the rebuttal deadline is approaching, we kindly request a prompt response based on the updated paper content and ask for a reconsideration of your evaluation score. Thank you for your support!

---

### Note · Authors · 2024-12-14

I have read and agree with the venue's withdrawal policy on behalf of myself and my co-authors.